# Machine learning methods to predict sea surface temperature and marine heatwave occurrence: a case study of the Mediterranean Sea

**Giulia Bonino**[1], **Giuliano Galimberti**[2], **Simona Masina**[1], **Ronan McAdam**[1], **and Emanuela Clementi**[1]

[1]CMCC Foundation – Euro-Mediterranean Center on Climate Change, Bologna, Italy
[2]Department of Statistical Sciences, University of Bologna, Bologna, Italy

**Correspondence:** Giulia Bonino (giulia.bonino@cmcc.it)

**Abstract.** Marine heatwaves (MHWs) have significant social and ecological impacts, necessitating the prediction of these extreme events to prevent and mitigate their negative consequences and provide valuable information to decision-makers about MHW-related risks. In this study, machine learning (ML) techniques are applied to predict sea surface temperature (SST) time series and marine heatwaves in 16 regions of the Mediterranean Sea. ML algorithms, including the random forest (RForest), long short-term memory (LSTM), and convolutional neural network (CNN), are used to create competitive predictive tools for SST. The ML models are designed to forecast SST and MHWs up to 7 d ahead. For each region, we performed 15 different experiments for ML techniques, progressively sliding the training and the testing period window of 4 years from 1981 to 2017. Alongside SST, other relevant atmospheric variables are utilized as potential predictors of MHWs. Datasets from the European Space Agency Climate Change Initiative (ESA CCI SST) v2.1 and the European Centre for Medium-Range Weather Forecasts (ECMWF) ERA5 reanalysis from 1981 to 2021 are used to train and test the ML techniques. For each area, the results show that all the ML methods performed with minimum root mean square errors (RMSEs) of about 0.1 °C at a 1 d lead time and maximum values of about 0.8 °C at a 7 d lead time. In all regions, both the RForest and LSTM consistently outperformed the CNN model across all lead times. LSTM has the highest predictive skill in 11 regions at all lead times. Importantly, the ML techniques show results similar to the dynamical Copernicus Mediterranean Forecasting System (MedFS) for both SST and MHW forecasts, especially in the early forecast days. For MHW forecasting, ML methods compare favorably with MedFS up to 3 d lead time in 14 regions, while MedFS shows superior skill at 5 d lead time in 9 out of 16 regions. All methods predict the occurrence of MHWs with a confidence level greater than 50 % in each region. Additionally, the study highlights the importance of incoming solar radiation as a significant predictor of SST variability along with SST itself.

## 1 Introduction

Accurate predictions of sea surface temperature (SST) and its extremes are important for many aspects of modern society. Anticipated changes include a rise in the occurrence and severity of prolonged sea surface temperature extremes lasting a minimum of 5 d, commonly known as marine heatwaves (MHWs; Hobday et al., 2016). These shifts have the potential to exert greater pressure on marine organisms and ecosystems, testing the boundaries of their adaptability and resilience (Frölicher and Laufkötter, 2018; Garrabou et al., 2022). MHWs can affect marine biodiversity (Garrabou et al., 2022; Cramer et al., 2018; Marbà et al., 2015; Rivetti et al., 2014; Juza et al., 2022) and the fishing and aquaculture industries (Cavole et al., 2016; Chandrapavan et al., 2019). Thus, SST prediction, and in turn MHW prediction, can support a range of adaptive and management activities for the Mediterranean marine ecosystems.

Forecasting the anomalous oceanic and atmospheric patterns that drive the SST variability in the build-up to these extreme events is still a challenge (Jacox et al., 2022; Hol-

brook et al., 2020). In the last decades, dynamical ocean forecasting systems have increased spatial resolution and improved data assimilation techniques, simulating the dynamics of the global ocean down to a few kilometers, with the aim of kilometric resolution in their future generations (Leroux et al., 2022). Dynamical ocean predictions have reached a remarkable degree of reliability, although the required computational resources are enormous (Alvarez Fanjul et al., 2022). In recent years, increasing interest has been given to machine learning (ML) techniques, even though, in contrast to the dynamical model, ML techniques do not know when they are violating the laws of physics (Buizza et al., 2022). As a "learning from data" approach, machine learning has the advantages of computational efficiency, accuracy, transferability, flexibility, and ease-of-use in ocean forecasting studies (Boukabara et al., 2019; Li et al., 2020; Wei and Guan, 2022; Taylor and Feng, 2022). Moreover, they are also less prone to model bias errors (Jacox et al., 2020), and, beyond computational efficiency, ML techniques excel in approximating nonlinear functions (Hornik, 1991).

Therefore, machine learning provides new opportunities for SST prediction (Boukabara et al., 2019). In contexts such as fishing, sporting events, coral bleaching, and aquaculture, the SST prediction can be treated as a time-series regression problem, where SST prediction is either restricted to a few locations or applied to SST values averaged over a region (Haghbin et al., 2021). Machine learning techniques include both shallow methods, such as linear regression and random forest (RForest) models, and deep learning models, such as artificial neural network (ANN), recurrent neural network (RNN), and convolutional neural network (CNN) models. As widely reviewed by Haghbin et al. (2021), linear regression and statistical methods have historically been extensively applied to SST estimation (Anding and Kauth, 1970; Corchado, 1995; McMillin, 1975). Pioneering works on SST prediction using deep learning methods are, instead, more recent. In the last few years, as reported in a recent review paper (Haghbin et al., 2021), deep-learning-based models such as the RNN, in particular the long short-term memory (LSTM; e.g., Xiao et al., 2019; Liu et al., 2018; Xie et al., 2019, and CNN (e.g., Han et al., 2019) have attracted progressively more attention in the research community, providing accurate estimates among the models considered. It is important to note that within the methods explored in this paper, LSTM stands out as the sole technique explicitly crafted for handling time-series data. In contrast, CNN algorithms are commonly tailored for image processing tasks and are not inherently designed for time-series forecasting.

To the best of our knowledge, just a few studies employed the RForest model to predict sea surface temperature (Wolff et al., 2020), but it has recently been successfully employed by Giamalaki et al. (2022) to directly predict marine extremes. Other attempts to predict extremes using ML techniques have been proposed by Chattopadhyay et al. (2020) and Jacques-Dumas et al. (2022) for land-based heatwaves

over North America and France, respectively. Nevertheless, the advantage of having the SST prediction, instead of the MHW events, is that end users or management operators could establish thresholds based on their needs (Jacox et al., 2022).

In short, given the impacts of MHWs on ocean ecosystems and the resulting economic losses of marine industries, there is an increasing need for MHW forecasts to help ocean users to be prepared for these events. Short-duration extreme marine heatwaves (MHWs), lasting only a week, can exert sudden stresses on temperature-sensitive aquatic life stages, potentially leading to immediate adverse effects such as mass mortality, disease, large-scale coral bleaching, and reduced seagrass meadows (Holbrook et al., 2020). For proactive marine management, operators in the coastal and marine sectors (e.g., fisheries and aquaculture and coastal water management) can use MHW event predictions for better planning of their activities. Here, we provide a proof-of-concept study on the advantage of data-driven ML methods to forecast the evolution of the SST state and its extremes in Mediterranean Sea regions up to 7 d ahead. The ML methods are compared against the Copernicus Mediterranean Forecasting System (MedFS; i.e., a dynamical ocean model, Clementi et al., 2021). MedFS is a numerical ocean prediction system, implemented and developed by the Euro-Mediterranean Center on Climate Change (CMCC), that produces analyses and short-term forecasts for the entire Mediterranean Sea and adjacent areas in the Atlantic Ocean (Clementi et al., 2021). The Mediterranean Sea is a well-studied hot spot for MHW events (Garrabou et al., 2009; Giorgi, 2006; Cramer et al., 2018; Pastor et al., 2020; Pastor and Khodayar, 2022; Garrabou et al., 2022; Ciappa, 2022). As detailed in the study by Bonino et al. (2023b), the Adriatic Sea, the Gulf of Lion, and the Alboran Sea encountered the highest occurrence of MHWs, the shortest-duration MHWs, and the most severe MHWs. The Mediterranean Sea serves as our study area due to its relevance for marine management activities; indeed more than 95 % of the global production of sea bream and sea bass comes from aquaculture, of which 97 % is produced by Mediterranean countries (Carvalho and Guillen, 2021). To the best of our knowledge, this is one of the first attempts to predict SST and, in turn, MHWs in the Mediterranean Sea 1 week ahead using machine learning techniques.

This paper is organized as follows: in Sect. 2, we describe the methodological framework to build the machine learning techniques to predict SST and, in turn, MHWs. Section 3 reports the results and the comparison with a dynamical model. Our conclusions and outlook of the work are summarized in Sect. 4.

## 2 Methodological framework

In this comprehensive study, the focus is on predicting SST time series and MHWs in 16 regions of the Mediterranean

Sea (Fig. 1) using ML techniques. In the following section we present the workflow of this study, which is summarized in Fig. 2.

## 2.1 Data collection and preprocessing

The machine learning techniques are trained, tested, and validated using the European Space Agency (ESA) Climate Change Initiative SST dataset v2.1, referred to as the ESA CCI SST dataset in the following text (Merchant et al., 2019). This dataset, accessible through the CEDA catalogue (https://catalogue.ceda.ac.uk/uuid/62c0f97b1eac4e0197a674870afe1ee6, last access: 20 MArch 2024), offers global daily satellite-derived sea surface temperature (SST) data. This dataset consists of daily maps of average SST at 20 cm nominal depth with 0.05 × 0.05° of horizontal resolution, covering the period from September 1981 to December 2016. Additionally, to expand the temporal coverage, we incorporate daily sea surface temperature data from 2017 to 2021, available via the Copernicus Climate Data Store (CDS) (https://cds.climate.copernicus.eu/cdsapp#!/dataset/satellite-sea-surfacetemperature?tab=overview, last access: 20 March 2024). The extended data from 2017 to 2021 are generated at level 4 (L4) by the Copernicus Climate Change Service (C3S), building upon the foundation of the ESA CCI SST dataset. These datasets are derived using software and algorithms developed within the framework of the ESA CCI SST project. For a comprehensive insight into the updates in processing for the ESA CCI SST dataset v2.1, detailed information is provided in the work by Merchant et al. (2019). The relevant atmospheric variables (AtmVs) are taken from the European Centre for Medium-Range Weather Forecasts (ECMWF) ERA5 dataset (Hersbach et al., 2020). Specifically, we select sea level pressure (SLP), geopotential height at 500 hPa (GEO), wind speed (WS), sensible heat flux (SENS), latent heat flux (LAT) and incoming solar radiation (INC) as the sum of the short- and longwave radiation downwards (i.e., into the ocean). SST and AtmVs are averaged over the Mediterranean regions (Fig. 1) to obtain time series of SST and AtmVs from 1981 to 2021. Moreover, we also consider the months of the year (MM) as an input variable, in order to describe the seasonality. Before building SST prediction tools based on machine learning, we analyze the mutual information between SST, SST itself, and AtmVs at different time lags (i.e., days) to have insights into the most relevant variables that contribute to the SST prediction. Mutual information between two random variables assesses their interrelationship. It signifies the amount of information that can be gained from one variable by observing another. A higher mutual information value corresponds to a greater reduction in uncertainty. Conversely, when mutual information is zero, the two variables are considered independent and unrelated. The mutual information of variable $X$ and variable $Y$ is defined as

$$\mathrm{MI}(x, y) = \int\limits_{x} \int\limits_{y} p_{XY}(x, y) \log \frac{p_{X,Y}(x, y)}{p_X(x) p_Y(y)}, \tag{1}$$

TS1 where $p_{XY}(x, y)$ is the joint probability density of $X$ and $Y$, and $p_X(x)$ and $p_Y(y)$ are the marginal probability density of $X$ and $Y$, respectively.

## 2.2 Machine learning techniques

Here, we briefly introduce the three ML techniques used in this study: a long short-term memory network (LSTM), a one dimensional convolutional neural network (CNN), and a random forest model (RForest). While we offer a concise overview of their architectures here, for comprehensive details, readers can refer to Breiman (2001) for RForest and Haghbin et al. (2021) for LSTM and CNN. All these ML models are multifaceted and multivariate, projecting SST for seven subsequent time steps by utilizing various input variables. The input sequences encompass SST data and the specified atmospheric variables (AtmVs) detailed in the preceding section, spanning the previous 7 d. To ensure comparability across diverse atmospheric and oceanic variables, we normalize these data using a min–max scaler. These ML frameworks receive concatenated extensive vectors structured in the format [time steps, lags, variables]. Here, "lags" denote the previous 7 d, while "variables" encompass SST and the predictors. We develop LSTM and CNN architectures using Keras high-level API of the TensorFlow platform built in Python (https://www.tensorflow.org/api_docs/python/tf/keras, last access: 20 March 2024) and random forest models using the RandomForestRegressor function of the sklean package of Python (https://scikit-learn.org/stable/modules/generated/sklearn.ensemble.RandomForestRegressor.html, last access: 20 March 2024).

- *Long short-term memory.* LSTM networks are types of recurrent neural networks capable of learning order dependence in sequence prediction problems, and they have been widely applied in temperature forecasting problems (Haghbin et al., 2021; Tran et al., 2021; Guo et al., 2022). We define the LSTM with 60 neurons with a hyperbolic tangent activation function in the first and unique hidden layer and 7 neurons in the dense layer (i.e., output layer) for predicting SST. Mean square error is used as the loss function. The network was trained for 200 epochs using the Adam optimizer with a learning rate of 0.0001 and batch size of 150.

- *Convolutional neural network.* CNNs have gained significant popularity in domains like image processing and computer vision. In recent times, there has been a noticeable surge in interest within the research community to use CNNs for solving time-series forecasting

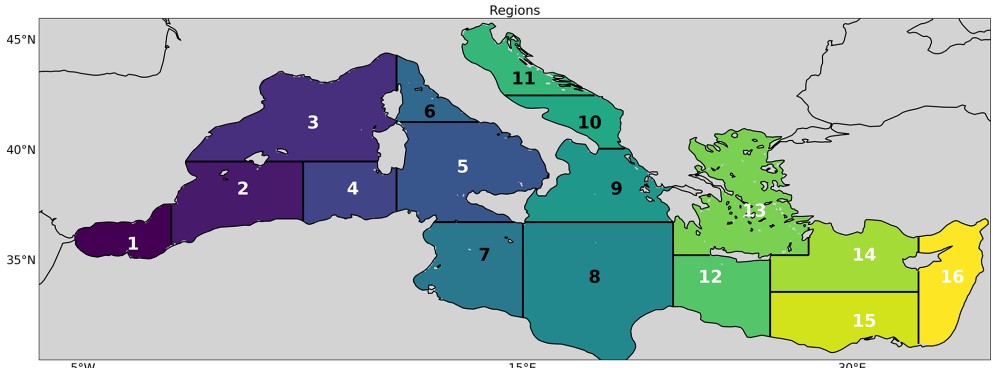

**Figure 1.** Mediterranean Sea regional subdivision and corresponding indices.

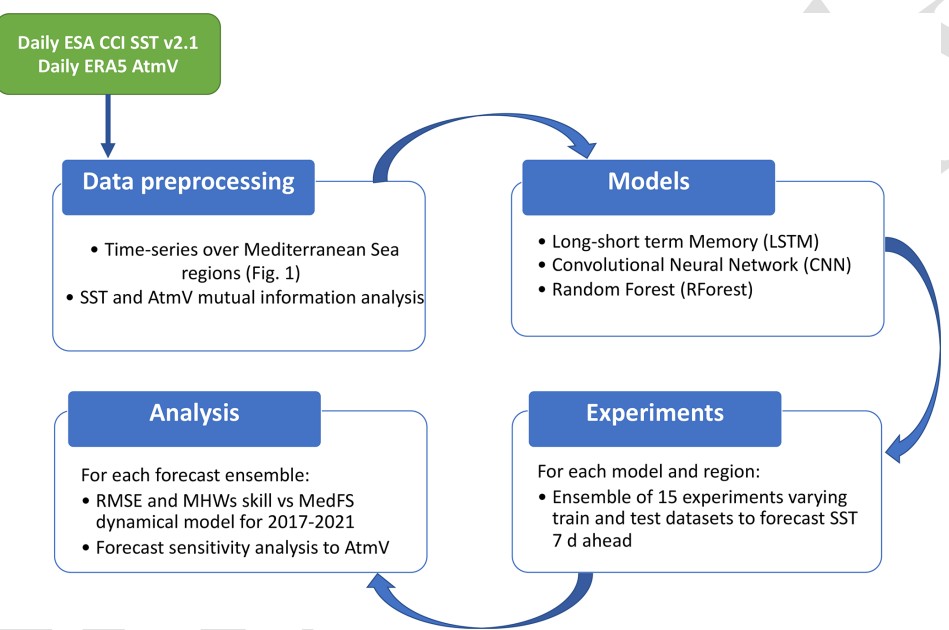

**Figure 2.** Flow diagram used in this study.

problems. We define the one-dimensional CNN with 64 channels with a kernel size of 2 and with a rectified linear activation function in the first and unique hidden layer. It is followed by a maxpool layer which divides the data size by 2. Finally a flatten operation and seven neurons in the output layer followed. Mean square error is used as the loss function. The network was trained for 200 epochs using the Adam optimizer with a learning rate of 0.0001 and batch size of 150.

– *Random forest model.* RForest is an ensemble learning method for classification and regression tasks that operates by constructing a multitude of randomly perturbed decision trees starting from the same train set. For regression tasks, the mean or average prediction of the individual trees is returned. We design the RForest model with 100 decision trees and 42 predictors randomly se-

lected to perform each split to construct decision trees. The function to measure the quality of a split during training is the mean squared error.

Notably, in our methodology, we opted not to incorporate dropout layers within the network architecture. Despite the absence of dropout regularization, our training results did not exhibit overfitting tendencies, suggesting a congruence between the model's capacity and the dataset characteristics.

### 2.3   Experiments

A period of 36 years of the daily data of SST and atmospheric variables is used to train and validate the techniques, while the remaining 4 years is used to test it. Based on the years used to train and test the methods, we distinguish two kinds of experiments:

– *Reference experiments (REXPs)*. ML techniques are trained using 1981–2016 as the training period and 2017–2021 as the testing period. We are interested in predicting the future SST, so the methods have to learn the correct time evolution of the SST, and, moreover, we also want to compare the skill with the MedFS dynamical model. MedFS predictions, part of the Copernicus Marine Service since 2017, offer SST forecasts for lead times up to 9 d averaged across specified regions. Since the accuracy of the ML forecast decreases almost linearly with time, we decided to limit the comparison with MedFS-forecasted SST averages for lead times of 1, 3, and 5 d.

– *Uncertainty experiments (UEXPs)*. The sampling uncertainty of each ML technique is estimated by progressively sliding the training and the testing period window of 4 years from 1981 to 2017. This means that, for each region, we perform 14 uncertainty experiments. For example, the first experiment uses 1985–2021 as the training period and 1981–1984 as the testing period, while the second one merges 1981–1984 and 1989–2021 data together to create the training dataset, and it uses 1984–1988 data as the testing dataset and so on.

In addition, we investigate the role of each driver in affecting prediction skills. Thus, for each ML technique and for each experiment (REXPs and UEXPs), the prediction accuracy on the test set was evaluated after randomly permuting (shuffling) the value of each driver, one at time (i.e., one "experiment" for each shuffled driver). This random permutation is aimed at removing any information about SST conveyed by the drivers (that is, annihilating the mutual information between SST and the driver), thus voiding their contribution in predicting SST (Breiman, 2001). In the case of informative drivers, the shuffling is expected to severely affect prediction accuracy, while it should have a negligible impact when considering uninformative drivers. In the following we refer to these sensitivity experiments as "SEXPs".

The computational time required for the ML experiments is about 2 min to train the method and 30 s to test it, while MedFS requires 10 min to simulate 1 d (i.e., 70 min for 7 d of forecast). We run all the experiments on the CMCC supercomputer ZEUS, which comprises 348 Lenovo SD530 biprocessor nodes (totaling 12 528 cores) interconnected via an Infiniband EDR network and boasts a total theoretical computing power of 1202 TFlops.

## 2.4 Evaluation metrics

For evaluating the ML-based prediction skills, we use a metric that is commonly applied in the SST forecast domain. In particular, we calculate the root mean square error (RMSE; Eq. 2) of the predicted SST in the test datasets against the ESA CCI SST dataset (i.e., observed SST). The RMSE measures the mean squared distance between the daily predicted

($F_i$) and the daily actual ($T_i$) SST in the $N$ samples of the test dataset. The RMSE is negatively oriented with a perfect value of 0.

$$\mathrm{RMSE} = \sqrt{\frac{1}{N} \Sigma_{i=1}^{n} (T_i - F_i)^2} \qquad (2)$$

Moreover, we also assess the ML techniques' accuracy in predicting MHW occurrence by detecting MHWs in the test time series and in the predicted time series. Notably, the test time series represents the regional mean SST, resulting in a single time series per region. MHWs are defined in each regional time series as in Hobday et al. (2016): SST higher, for 5 d or longer, than the 90th percentile threshold of seasonally varying climatology calculated over more than a 30-year period without removing the long-term trend. If an event occurs less than 2 d before the end of another, it is considered part of the ongoing MHW. An 11 d sliding window centered on each day is used for the climatology calculation. The climatology and the threshold are calculated over the reference training period (i.e., 1982–2016) and applied to the testing period (i.e., 2017–2021). Even though MHWs are defined considering at least 5 d of consecutive anomalies, the differences between the predicted and observed datasets are calculated day by day. The detection performance is assessed by computing, in the test set, the false positive rate (FPR; Type I error: incorrect detection of MHW) and false negative rate (FNR; Type II error: non-detection when MHW occurs) and the $F_1$ score. FPR and FNR are defined as

$$\mathrm{FPR} = \frac{\mathrm{FP}}{\mathrm{FP} + \mathrm{TN}} \quad \text{and} \quad \mathrm{FNR} = \frac{\mathrm{FN}}{\mathrm{FN} + \mathrm{TP}}, \qquad (3)$$

where FP is the number of false positives, TN is the number of true negatives, FN is the number of false negatives, and TP is the number of true positives (see Table 2). Note that the true positive rate (TPR) and the true negative rate (TNR) are $\mathrm{TPR} = 1 - \mathrm{FNR}$ and $\mathrm{TNR} = 1 - \mathrm{FPR}$, respectively. The $F_1$ score is a single overall measure of prediction accuracy (Eq. 4) and takes into account the imbalance of the dataset: around 1/7 of the test samples are MHW events. It is calculated from the precision and recall scores.

$$F_1 = 2 \times \frac{\mathrm{precision} \times \mathrm{recall}}{\mathrm{precision} + \mathrm{recall}} \quad \text{where}$$

$$\mathrm{precision} = \frac{\mathrm{TP}}{(\mathrm{TP} + \mathrm{FP})} \quad \text{and}$$

$$\mathrm{recall} = \frac{\mathrm{TP}}{(\mathrm{TP} + \mathrm{FN})} \qquad (4)$$

We also assess the mean forecasted intensity ($\mathrm{Imean_{ML}}$) within the REXPs for each ML technique and compare it to the observed intensity mean from the ESA CCI dataset ($\mathrm{Imean_{ESA}}$), computing the intensity error (IE), formulated as $\mathrm{IE} = \mathrm{Imean_{ESA}} - \mathrm{Imean_{ML}}$. Furthermore, we compare the performance of the SST and MHW prediction of the REXPs against MedFS SST forecast data.

**Table 1.** Mutual Information between SST and AtmVs not lagged in time (LAG0) and with 7 d lag (LAG7). The lag time is expressed in days. Bold values identify the highest mutual information values for LAG0 and LAG7.

| Region ID | WIND LAG0 | WIND LAG7 | GEO LAG0 | GEO LAG7 | SLP LAG0 | SLP LAG7 | LAT LAG0 | LAT LAG7 | SENS LAG0 | SENS LAG7 | INC LAG0 | INC LAG7 | MM LAG0 | MM LAG7 |
|---|---|---|---|---|---|---|---|---|---|---|---|---|---|---|
| 1 | 0.03 | 0.03 | 0.47 | 0.38 | 0.14 | 0.14 | 0.03 | 0.02 | 0.1 | 0.08 | **0.64** | **0.55** | 0.25 | 0.21 |
| 2 | 0.01 | 0.04 | 0.46 | 0.39 | 0.14 | 0.12 | 0.06 | 0.03 | 0.12 | 0.10 | **0.71** | **0.63** | 0.25 | 0.21 |
| 3 | 0.08 | 0.04 | 0.46 | 0.36 | 0.13 | 0.1 | 0.03 | 0.03 | 0.11 | 0.08 | **0.75** | **0.62** | 0.28 | 0.23 |
| 4 | 0.05 | 0.06 | 0.48 | 0.41 | 0.11 | 0.11 | 0.03 | 0.05 | 0.07 | 0.07 | **0.68** | **0.60** | 0.25 | 0.21 |
| 5 | 0.08 | 0.06 | 0.52 | 0.42 | 0.12 | 0.11 | 0.05 | 0.06 | 0.08 | 0.07 | **0.79** | **0.68** | 0.26 | 0.21 |
| 6 | 0.06 | 0.05 | 0.46 | 0.38 | 0.12 | 0.10 | 0.05 | 0.05 | 0.08 | 0.10 | **0.62** | **0.75** | 0.26 | 0.31 |
| 7 | 0.07 | 0.05 | 0.48 | 0.41 | 0.12 | 0.10 | 0.06 | 0.06 | 0.06 | 0.06 | **0.81** | **0.73** | 0.21 | 0.17 |
| 8 | 0.07 | 0.06 | 0.49 | 0.43 | 0.13 | 0.10 | 0.06 | 0.08 | 0.12 | 0.12 | **0.87** | **0.79** | 0.22 | 0.18 |
| 9 | 0.06 | 0.06 | 0.53 | 0.45 | 0.14 | 0.12 | 0.03 | 0.05 | 0.09 | 0.07 | **0.78** | **0.67** | 0.26 | 0.22 |
| 10 | 0.05 | 0.04 | 0.53 | 0.43 | 0.13 | 0.12 | 0.04 | 0.04 | 0.10 | 0.10 | **0.73** | **0.62** | 0.31 | 0.26 |
| 11 | 0.06 | 0.04 | 0.47 | 0.41 | 0.14 | 0.12 | 0.05 | 0.08 | 0.11 | 0.10 | **0.79** | **0.66** | 0.36 | 0.31 |
| 12 | 0.05 | 0.05 | 0.49 | 0.41 | 0.16 | 0.11 | 0.06 | 0.05 | 0.10 | 0.08 | **0.82** | **0.73** | 0.22 | 0.18 |
| 13 | 0.04 | 0.05 | 0.56 | 0.48 | 0.17 | 0.13 | 0.05 | 0.05 | 0.12 | 0.09 | **0.77** | **0.64** | 0.29 | 0.25 |
| 14 | 0.05 | 0.06 | 0.57 | 0.46 | 0.23 | 0.18 | 0.06 | 0.08 | 0.10 | 0.07 | **0.83** | **0.72** | 0.24 | 0.21 |
| 15 | 0.05 | 0.05 | 0.51 | 0.42 | 0.20 | 0.18 | 0.06 | 0.09 | 0.10 | 0.09 | **0.84** | **0.74** | 0.22 | 0.19 |
| 16 | 0.06 | 0.04 | 0.59 | 0.48 | 0.26 | 0.22 | 0.08 | 0.09 | 0.18 | 0.17 | **0.85** | **0.74** | 0.23 | 0.20 |

**Table 2.** Accuracy in predicting MHWs.

| | MHW not predicted | MHW predicted |
|---|---|---|
| MHW not observed | TN | FP |
| MH predicted | FN | TP |

## 3 Results

In the following, we evaluate the predictive skill of the ML techniques for SST predictions and, in turn, MHW predictions, up to a lead time of 7 d. For the presented results, the statistical index used to evaluate the performances is the RMSE for SST and the $F_1$ scores for MHWs (see Sect. 2.4., "Evaluation metrics"). Moreover, we also assess the sensitivity of the ML techniques with respect to the input variables in the testing dataset.

### 3.1 Mutual information analysis

Table 1 shows the mutual information between SST and the selected variables not lagged in time (LAG0) and with a 7 d lag (LAG7). In all the regions and for all the variables (except some rare cases for LAT), the mutual information decreases when increasing the time lag. In all the regions and for both the time lags, the INC and then the GEO are the variables that seem to show the strongest association with SST, with values that range from 0.55 to 0.87 and from 0.38 to 0.59, respectively. INC thermally influences the temperature along the water column, while GEO impacts ocean variability as a proxy for the large-scale atmospheric circulation. Increased GEO values correspond to higher air pressure above the ocean, resulting in warmer SST conditions. Progressively, we find MM, SLP, and SENS with mean values of about 0.20, 0.15, and 0.10, respectively. WIND and LAT show low values, usually less than 0.1. However, it is worth noting that these values are substantially lower than those obtained considering the dependence between SST and itself at different lag (see Fig. S1 in the Supplement, reporting the mutual information heatmaps for all the time lags in each region). Despite the aforementioned values, reduced air–sea heat fluxes, in particular the latent heat, and reduced wind speed have been associated with MHWs in the mid-latitudes (Vogt et al., 2022). Therefore, we decide to retain all the selected AtmVs as potential predictors of SST variability over the Mediterranean Sea.

### 3.2 SST prediction

To have an overall insight into the methods' performances over the Mediterranean Sea, we first examine the ML methods' performance of the daily SST prediction in all the regions for the REXPs (Fig. 3). Overall, these methods show similar ranges of RMSE; they display minimum values of about 0.1 °C at lead time of 1 d (L1) and maximum values of about 0.8 °C at lead time of 7 d (L7). The RMSEs grow with the increasing forecast lead time, and the evolution of the RMSE in each region is consistent across ML techniques. For instance, in all the techniques, region 15 shows the lowest, or almost the lowest (e.g in CNN) RMSE, while region 11 shows the highest errors. To visualize the errors spatially, Fig. 4 shows the mean RMSE for LSTM for each region (CNN and RForest results are shown in Fig. S2). The areas with larger errors are as follows: (i) the Alboran Sea, strongly characterized by a complex dynamics influenced by the incoming cold Atlantic water through the Gibraltar Strait

**Table 3.** ML network performance for the SST daily predictions in terms of root mean square error (RMSE) for long short-term memory (LSTM), random forest (RForest), and convolutional neural network (CNN) on the first day of forecast (forecast lead 1, L1) and the seventh days of forecast (forecast lead 7, L7). Bold values identify the best performance (i.e., lowest RMSE) for L1 and L7.

| | LSTM | | RForest | | CNN | |
|---|---|---|---|---|---|---|
| Region ID | L1 | L7 | L1 | L7 | L1 | L7 |
| 1 | **0.17** | **0.59** | 0.21 | 0.62 | 0.23 | 0.62 |
| 2 | **0.15** | **0.45** | **0.15** | **0.45** | 0.19 | 0.48 |
| 3 | **0.16** | **0.70** | 0.18 | 0.73 | 0.23 | 0.76 |
| 4 | 0.14 | **0.45** | **0.16** | 0.50 | 0.29 | 0.47 |
| 5 | **0.13** | **0.58** | 0.16 | **0.58** | 0.21 | 0.63 |
| 6 | **0.19** | **0.65** | 0.20 | 0.71 | 0.25 | 0.69 |
| 7 | **0.14** | **0.43** | 0.16 | 0.47 | 0.19 | 0.47 |
| 8 | 0.14 | **0.39** | **0.13** | 0.40 | 0.22 | 0.40 |
| 9 | **0.14** | **0.57** | 0.17 | 0.59 | 0.31 | 0.64 |
| 10 | **0.16** | **0.63** | 0.19 | 0.71 | 0.27 | 0.72 |
| 11 | **0.19** | **0.74** | 0.22 | 0.78 | 0.28 | 0.80 |
| 12 | **0.12** | **0.38** | 0.13 | 0.39 | 0.29 | 0.39 |
| 13 | **0.13** | **0.50** | 0.15 | **0.50** | 0.18 | 0.52 |
| 14 | **0.12** | **0.44** | 0.14 | 0.46 | 0.20 | 0.47 |
| 15 | **0.11** | **0.35** | 0.13 | **0.35** | 0.20 | 0.36 |
| 16 | 0.16 | 0.41 | **0.14** | **0.38** | 0.21 | 0.38 |

modulating the water transport; (ii) the northwest part of the basin, which is an area of dense water formation and intense dynamics due to the Gulf of Lion gyre (Madec et al., 1991; Pinardi et al., 2006) and a boundary intense current called ₅ the Liguro–Provençal–Catalan Current (Pinardi et al., 2006); (iii) the Adriatic Sea, especially in its northern shelf, characterized by a complex topography, intense air–sea exchanges, and large riverine inputs that contribute to enrich the dynamics of the area.

₁₀ Looking into the methods' performance comparison in more detail (Table 3), LSTM, followed by RForest, outperforms CNN at L1 and at L7. In particular at L1, LSTM has the highest predictive skill in 13 out of 16 regions and RForest in 2 out of 16 regions. In the remaining region, they score equally. For L7, LSTM has the highest predictive skill in 11 ₁₅ out of 16 regions and RForest in 1 out of 16 regions. In the remaining 4 regions, they score equally.

We select regions 11, 15, and 4 which display the highest, the lowest, and the intermediate RMSEs, respectively. We re- ₂₀ fer to region 4 as the "Western Mediterranean" (WM), to region 11 as the "Central Mediterranean" (CM), and to region 15 as the "Eastern Mediterranean" (EM). Figure 4a shows the RMSEs of the predicted SST by REXPs (solid line) and by UEXPs (bars) against the observed SST time series. More- ₂₅ over, the RMSEs of the SST predicted by MedFS (i.e., dynamical model) for the reference period (i.e., 2017–2021) are also reported. It is worth noting that the RMSE of the dynamical model does not show significant increases with the

forecast lead time, unlike ML techniques. Results for all the other regions are shown in Fig. S3. Referring to Figs. 5a and ₃₀ S3, we can appreciate that most of the ML techniques errors compare favorably with respect to the MedFS errors during the first days of forecast. In particular, the CM RMSEs range between a minimum error of about 0.19 °C at L1 and maximum error of about 0.74 °C at L7. All the ML methods show ₃₅ lower RMSE than MedFS for the first 3 d of forecast, and they are comparable at lead time of 5 d. EM and WM show lower variability of the error with respect to the CM, ranging in the intervals 0.11–0.35 °C and 0.14–0.45 °C, respectively. Over those two regions, it could be observed that all ₄₀ ML methods' skills are in line with the one of MedFS; they have similar RMSEs, CNN being the one showing higher error. In WM and CM – and in almost all the other regions (see Fig. S3) – the uncertainties tend towards higher RMSEs with respect to the REXP errors (i.e., errors represented by ₄₅ the solid line in Fig. 5a). It is likely connected to the fact that CNN algorithms are typically designed for image processing rather than time-series forecasting.

An additional analysis is presented to show how the different ML methods perform in predicting SST and MHW occur- ₅₀ rence at different forecast lead time. Figure 6 shows 1 year (2020) daily SST time series of the predicted and observed SST at L5 (Figs. S4 and S5 for L1 and L3, respectively) as well as the SST climatology, averaged in the three regions of interest. The figure shows a very close match between ₅₅ the forecasts and the observations, the SST variability being clearly well represented and forecasted by all the models (i.e., ML techniques and MedFS model). This is confirmed by the fact that the difference in annual means and standard deviations is minimal, generally within decimal values. ₆₀

## 3.3 MHW prediction

Going a step further in the prediction skill assessment, we also evaluate the ability of the different ML techniques in predicting MHWs' occurrence (Table 4 and Fig. 5b). To this end, we define MHWs using the method of Hobday et al. ₆₅ (2016) in the observed time series, in the predicted time series by REXPs and by MedFS (see Sect. 2.4., "Evaluation metrics", for more details). For the selected regions, Table 4 reports the false positive rate (FPR), the false negative rate (FNR), and the $F_1$ scores at L1, L3, and L5. Results for ₇₀ all the other regions are reported in Fig. S5 and Table S1. Overall, for all the forecast lead times and for all the ML techniques (except in rare cases; see Table S1), the FNR is higher than the FPR, meaning that the ML methods tend to underestimate SST peaks/extremes. The MedFS model, in- ₇₅ stead, shows mixed behavior: 7 out of 16 regions show higher FPR than FNR at all the lead times. In CM, MedFS shows a high FPR of about 27 % at L5 (Table 4) as it is also evident in Fig. 6. During January 2020 and 2021 the MedFS-predicted time series (blue line in Fig. 6) is usually greater ₈₀ than the 90th percentile threshold used to define MHWs

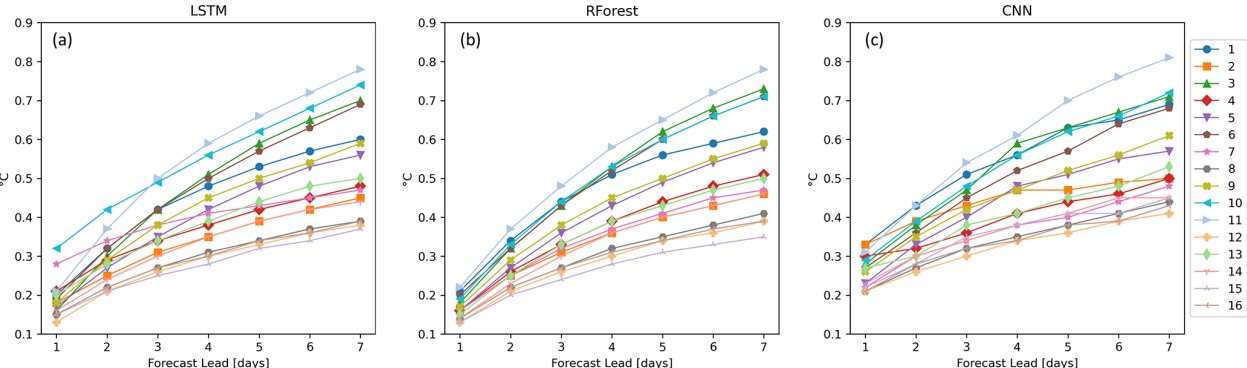

**Figure 3.** ML networks' performance for the SST daily predictions in terms of root mean square error (RMSE) in 16 regions of the Mediterranean Sea (different colors) for **(a)** long short-term memory networks, **(b)** random forest, and **(c)** convolutional neural networks.

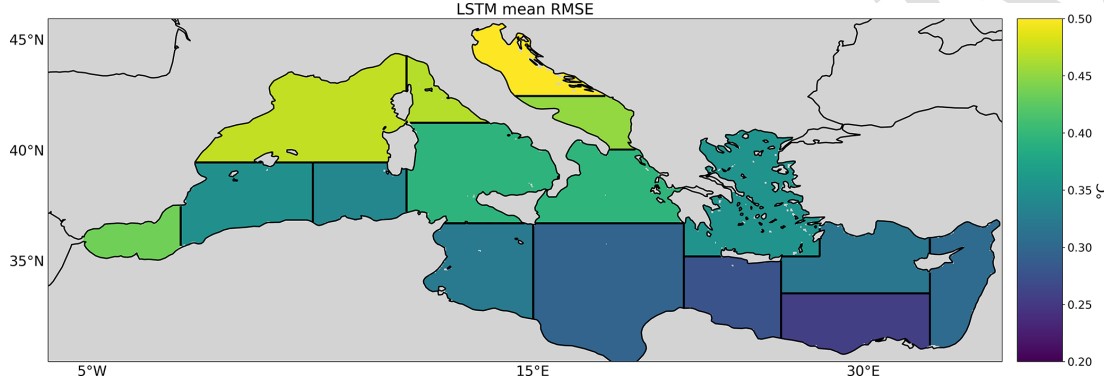

**Figure 4.** LSTM mean RMSE for REXP for each region.

(gray line in Fig. 6), leading to high FP (highlighted as circles in Fig. 6). The ML techniques show, instead, high FNR of about 50 % (highlighted as crosses in Fig. 6). Nevertheless, in WM, MedFS is in line with LSTM and RForest, showing 30 % of FNR. Looking from another perspective, and with the reminder that $TPR = 1 - FNR$, these rates demonstrate that, except for some rare cases, all the methods in all the regions predict the occurrence of a MHW with a confidence greater than 50 %.

To have a more accurate evaluation of the performance of the methods, instead of showing just the errors (i.e., FPR and FNR), we also evaluate the $F_1$ score (Fig. 5b and Tables 4 and S1). Note that the $F_1$ score ranges between 0 and 1 and it is positively oriented. As expected, the $F_1$ scores decrease when increasing the forecast lead time, but, at all the lead times, they show values greater than 0.5, meaning satisfactory MHW predictions for all the ML techniques in all the regions (except some rare cases, especially for CNN). Overall, comparing the performances of the ML techniques, the RForest and the LSTM outperform CNN in all the regions, at least on the first days of forecast. The $F_1$ score results show that LSTM outperforms in the CM and WM at all lead times, reaching a $F_1$ score of about 0.9 at L1 (Fig. 5b). EM shows different behavior: LSTM has the best predictive skill, outperformed by RForest in the following lead times. Nevertheless, as for the SST RMSE, all the ML techniques provide nearly same results; indeed, the ML differences in terms of $F_1$ score are usually around 0.15. Comparing the results with the MedFS model (blue circles in Fig. 6) we can appreciate that in the selected regions the ML techniques outperform MedFS up to L5 for CM and up to L3 for WM and EM (Table 4). It is worth noting that in all the other regions (except region 8), ML methods outperform MedFS up to L3, while at L5 in 9 out of 16 regions MedFS has the best skill (Table S1).

We also evaluate the differences in MHW intensity mean predicted by the ML models and MedFS against observation (i.e., intensity error, IE; see Sect. 2.4., "Evaluation metrics") during the studied period (Fig. 5c). For WM, CM, and EM it is interesting to note that MedFS shows positive IEs, meaning that the MedFS-predicted intensity is overestimated, while the ML models show opposite behavior, IEs are negative. This characteristic is evident also for the other regions (Fig. S6), except regions 1, 12, 13 where MedFS IEs are negative. It is also worth highlighting that, in all the regions, the RForest IEs, even if they are not the smallest ones, degrade slower than the LSTM IEs.

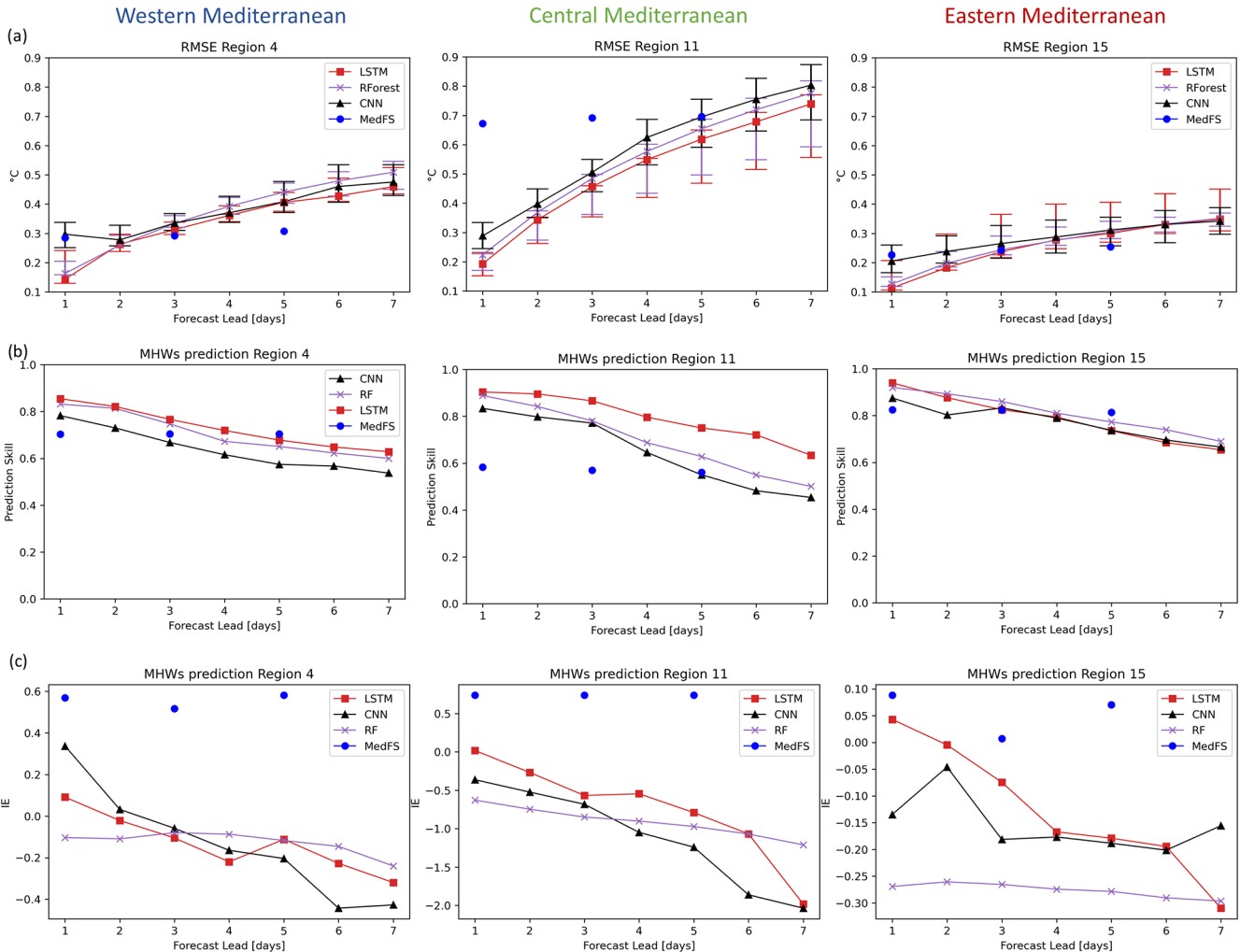

**Figure 5. (a)** ML network performance for the SST daily predictions in terms of root mean square error (RMSE) against MedFS performance and **(b)** variation of $F_1$ score for MHWs occurrence prediction with the forecast lead time and **(c)** variation of forecasted MHW intensity mean error (IE) with the forecast lead time for the (left column) Western Mediterranean, (middle column) Central Mediterranean, and (right column) Eastern Mediterranean. The sampling uncertainty of each prediction in **(a)** is illustrated by the bar. RMSE errors represented by the solid lines represent the reference experiments.

## 3.4 Sensitivity analysis

In this section, we discuss the results of the SEXPs (see Sect. 2.3., "Experiments", for details) for the three selected regions, noting that similar conclusions can be drawn for all the regions (not shown). The analysis thereby focuses on evaluating the methods' performance in terms of SST RMSE (Fig. 7). This means that the higher the increase in the RMSE after a driver is shuffled, the higher its predictive power. The labels of Fig. 7 indicate, for each experiment, the driver that has been shuffled in REXPs. For an easier interpretation of the results we are showing only the SST, the INC, and the LAT drivers, as they are the most relevant. For all the techniques, the RMSE increases notably with respect to the REXP when the SST is randomly modified; it grows

up to 6, 7, and 5.5 °C for WM, CM, and EM, respectively. Nevertheless, it is worth noting that the extent to which the RMSE increases after shuffling SST shows a tendency to decrease as the forecast lead time increases. This result suggests that the SST itself has the strongest predictive power in forecasting SST, slightly losing predictive skill increasing the lead times. The incoming solar radiation, to a lower extent, shows the opposite behavior: after shuffling, the RMSE tends to increase more than the other drivers with the forecast lead times. The RMSE at L7 reached values of about 1, 1.7, and 1 °C for WM, CM, and EM, respectively. Surprisingly, in contrast with the mutual information analysis, we can notice that for WM and EM the latent heat plays a role. In particular, for WM the RMSE at L7 reached values of about 0.5 for all the ML techniques, double the RMSE of REXPs. Overall,

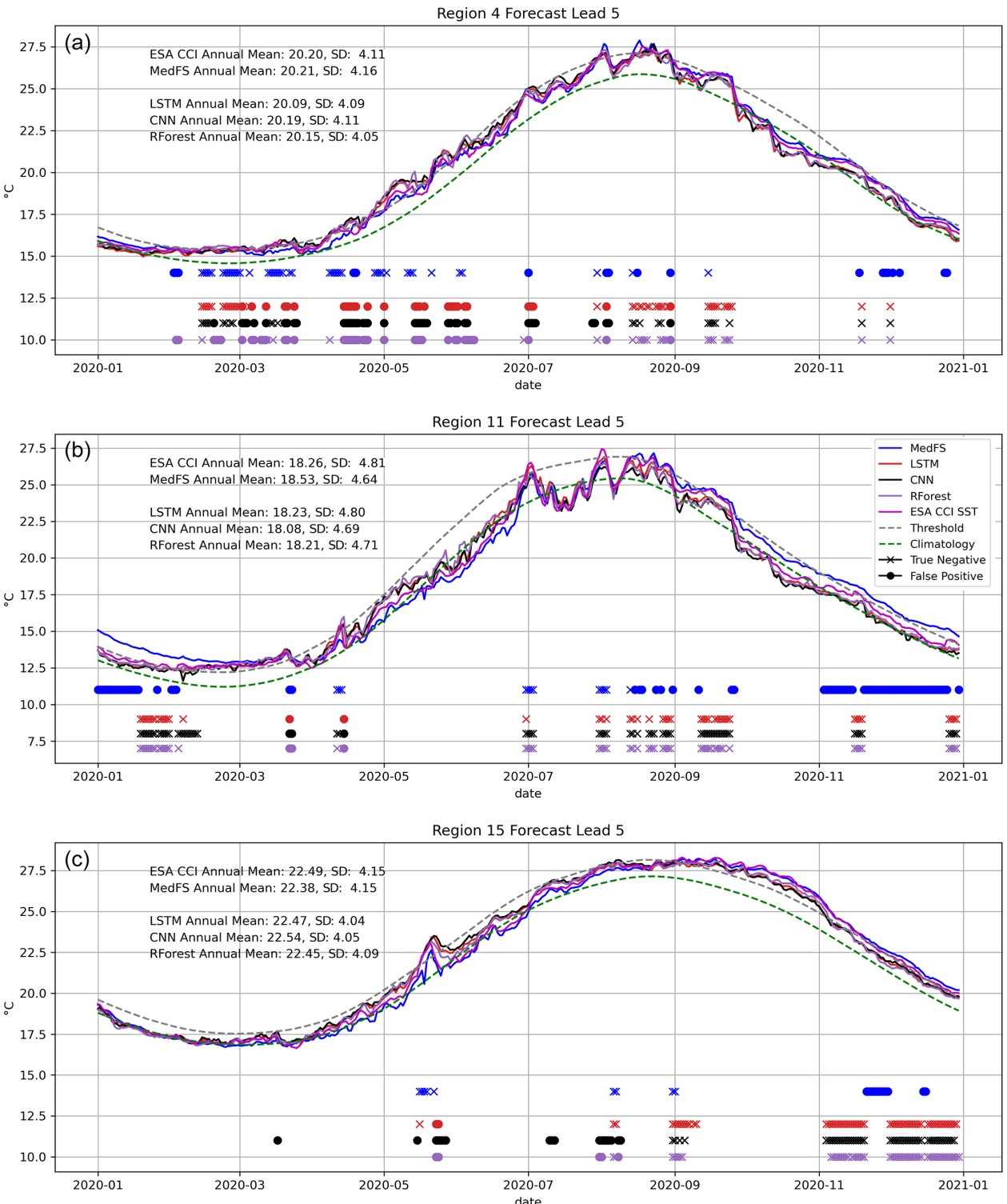

**Figure 6.** Time series of observed SST (ESA CCI SST) and predicted SST by the ML techniques (LSTM, CNN, RForest) and by MedFS at 5 d forecast lead time during 2020 for the **(a)** Western Mediterranean, **(b)** Central Mediterranean, and **(c)** Eastern Mediterranean. The 90th percentile threshold to define MHWs is represented in gray and the daily climatology in green. Crosses correspond to missed alarms (false negative) and points to false alarms (false positive) in the forecast output in predicting MHWs. Colors refer to the different ML techniques.

**Table 4.** ML networks' performance for the MHWs daily predictions in terms of false positive rate (FPR), false negative rate (FNR), and $F_1$ score for long short-term memory (LSTM), random forest (RForest), convolutional neural network (CNN), and the Mediterranean Forecasting System (MedFs) on the first day of forecast (forecast lead 1, L1), on the third day of forecast (forecast lead 3, L3), and on the fifth day of forecast (forecast lead 5, L5). Bold values identify the best $F_1$ scores at L1, L3, and L5. Stars (**) identify the highest rates of FPR and FNR.

| Region ID | Methods | FPR | | | FNR | | | $F_1$ score | | |
|---|---|---|---|---|---|---|---|---|---|---|
| | | L1 | L3 | L5 | L1 | L3 | L5 | L1 | L3 | L5 |
| 4-WM | LSTM | 0.04 | 0.04 | 0.07 | 0.13 | 0.33 | 0.39 | **0.86** | **0.78** | 0.71 |
| | RForest | 0.03 | 0.06 | 0.09 | 0.19 | 0.25 | 0.34 | 0.83 | 0.75 | 0.65 |
| | CNN | 0.07 | 0.04 | 0.05 | 0.17 | 0.42** | 0.51** | 0.78 | 0.67 | 0.57 |
| | MedFS | 0.07** | 0.07** | 0.06** | 0.29** | 0.31 | 0.31 | 0.70 | 0.70 | **0.72** |
| 11-CM | LSTM | 0.02 | 0.02 | 0.02 | 0.1 | 0.17 | 0.38 | **0.90** | **0.87** | **0.73** |
| | RForest | 0.03 | 0.02 | 0.030 | 0.12 | 0.30 | 0.49 | 0.89 | 0.78 | 0.63 |
| | CNN | 0.06 | 0.03 | 0.01 | 0.12** | 0.31** | 0.59** | 0.84 | 0.78 | 0.56 |
| | MedFS | 0.27** | 0.27** | 0.27** | 0.17 | 0.19 | 0.2 | 0.6 | 0.58 | 0.58 |
| 15-EM | LSTM | 0.03 | 0.05 | 0.1 | 0.06 | 0.21** | 0.29** | **0.94** | 0.84 | 0.74 |
| | RForest | 0.03 | 0.04 | 0.07 | 0.1 | 0.19 | 0.28 | 0.92 | **0.86** | 0.77 |
| | CNN | 0.15** | 0.12** | 0.15** | 0.02 | 0.09 | 0.25 | 0.86 | 0.85 | 0.73 |
| | MedFS | 0.13 | 0.11 | 0.12** | 0.12 | 0.15 | 0.16 | 0.82 | 0.82 | **0.81** |

the aforementioned analysis suggests that the incoming solar radiation, as shown also by the mutual information analysis, has some predictive power in driving SST variability. It is important to highlight that incoming solar radiation shows a tendency to gain predictive power as forecast leads increase, whereas SST, to a much lesser extent, tends to lose it. This suggests that atmospheric variables could be useful in forecasting SST at longer timescales. However, it is worth stressing that the ML methods look for statistical relations (e.g., linear or non-linear relations) between variables that do not necessarily have a physical meaning (e.g., a cause–effect relation).

## 4   Discussion

In this study, a group of ML algorithms – random forest (RForest), long short-term memory (LSTM), and convolutional neural networks (CNNs) – are used to evaluate their ability in building a competitive prediction tool of SST and MHW occurrence 7 d ahead in the Mediterranean Sea. The methods use the European Space Agency (ESA) Climate Change Initiative (CCI) sea surface temperature, sea level pressure (SLP), geopotential height at 500 hPa (GEO), wind speed (WS), sensible heat flux (SENS), latent heat flux (LAT), and incoming solar radiation (INC) from ECMWF ERA5 as input data. We compare the ML predictions against MedFS, part of the Copernicus Marine Service since 2017, which offers SST forecasts for lead times up to 9 d averaged across specified regions. It is important to underline that the data used in our work are designed primarily for climate studies and for providing a gap-free dataset through interpolation,

which raises concerns about potential biases introduced into our forecasting model. The interpolation process, while ensuring a comprehensive dataset, might inadvertently smooth variations and obscure critical phenomena like coastal upwelling. Alternative approaches utilizing near-real-time operational data could offer more dynamically responsive and less biased datasets for improved forecasting accuracy. Moreover, our methodology involves averaging SST data to obtain a single representative SST for each zone, and the same approach is used for atmospheric predictors. We acknowledge that averaging data in this manner might smooth out localized variations, thereby potentially overlooking non-linearity in the effects of variables such as wind speed. When data are averaged within an area, the issue tends to become more linear, allowing simpler methods like linear regression to yield good results (not shown). However, when working with higher-resolution data, ML methods outperform classical methods, demonstrating their potential and unearthing intricate nonlinear relationships.

A crucial aspect of this study is the comparison of the ML techniques' performance with that of the dynamical Copernicus Mediterranean Forecasting System (MedFS) for both SST and MHW forecasts. Impressively, ML methods demonstrated a favorable edge over MedFS, especially LSTM in the early forecast days. For MHW forecasting, ML methods outperformed MedFS in most regions up to 3 d of forecast lead time, while MedFS exhibited superior skill at 5 d of forecast lead time in 9 out of 16 regions. It is worth emphasizing that dynamical models, used to produce the 7 d forecast, are forced by atmospheric forecasts, enabling the ocean to be influenced by the overlying atmospheric conditions. Conversely, ML methods lack information about the atmospheric

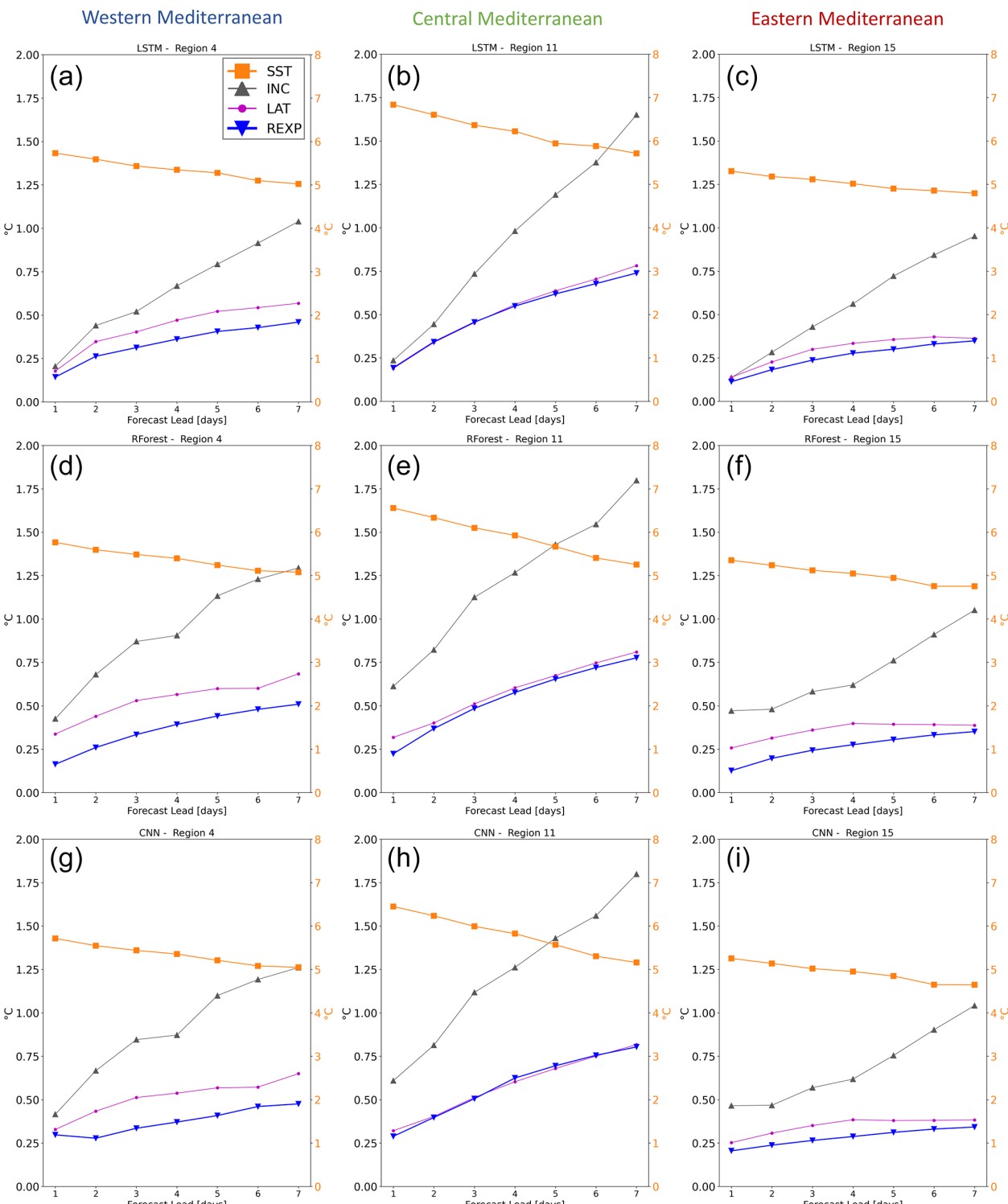

**Figure 7.** SST root mean square error (RMSE) of the sensitivity experiments (SEXPs) for each forecast lead time for the **(a, d, g)** Western Mediterranean, **(b, e, h)** Central Mediterranean, and **(c, f, i)** Eastern Mediterranean. The labels indicate, for each experiment, the driver that has been shuffled. The right *y* axis (orange ticks) refers to the SST driver experiment (solid orange line with "*x*" marker), and the left *y* axis indicates the other drivers' error.

conditions during the forecasted period and heavily depend on the autocorrelation structure of SST, which rapidly decreases as the time lag increases. These could be the causes of the faster increase in RMSE with the forecast lead times compared to MedFS. Results demonstrate comparable performance, at least in the earliest days of the forecast, to physics-based model simulations (i.e., Copernicus Mediterranean Forecasting System) but with the advantage of low computational cost. The low computational cost of these off-the-shelf ML tools has many advantages. First, the suite of methods presented here can be trained on a laptop and applied to any geographic location. Secondly, once trained, the ML techniques do not require high user skills to be correctly run and analyzed. Furthermore, they can be easily updated, once additional data become available. In addition, the advantage to having the SST prediction is that the end users could establish thresholds based on their needs. Marine users and stakeholders operating with different purposes and in different regions may need specific thresholds to define the extreme conditions which may limit their activities. However, the gap to dynamical models at 7 d of forecast lead time and the high rates of false negatives motivate future work to improve the performance of the underlying networks. For example, one may consider adding complexity or improving the model architectures. Passing from time-series forecasts to spatially complete maps of predicted sea surface temperatures (i.e., from 1D prediction to 2D prediction) is also appealing. These methods, contingent on higher computational time and resources, could be trained in each grid point of the target dataset, in order to obtain maps of SST prediction for each lead time at very high resolution.

Our findings also indicate that, in addition to SST itself (as also observed by Giamalaki et al., 2022), incoming solar radiation appears to play a role in predicting SST. These variables are inherently physically related, but it is important to note that ML techniques, unlike dynamical models, do not simulate the ocean's dynamics. Therefore, establishing a physical-process-based relationship between incoming solar radiation and MHW occurrences is premature. To comprehend this underlying connection, driver-based studies, such as those conducted by Holbrook et al. (2019), Schlegel et al. (2021), and Rodrigues et al. (2019), are necessary. Furthermore, the neural network algorithms applied lack inherent knowledge of physical laws, potentially leading to violations of fundamental physical constraints. This limitation likely restricts the proficiency of NNs in maintaining accuracy throughout the prediction interval (Boukabara et al., 2019; Dueben and Bauer, 2018). It prompts a need for cautious interpretation and validation of results, especially in scenarios where physical constraints significantly influence outcomes. While this study primarily focuses on predictive skill, future iterations could explore methodologies to infuse ML frameworks with physical laws governing ocean dynamics. For instance, Zanetta et al. (2023) propose achieving physical consistency in deep-learning-based postprocessing

models for temperature and humidity by incorporating meteorological expertise through analytic equations. Incorporating these constraints could potentially enhance the reliability of NN predictions, mitigating the risk of straying from physical realities. One may consider, due to the strengths of NNs in shorter-term predictions (3/5 d interval) and the potential limitations discussed, to use the ML model alongside models like MedFS for longer prediction intervals. Utilizing MedFS for longer-term forecasts could leverage its established reliability over extended periods, while ML could excel in shorter-term predictions.

Our study predominantly focuses on the influence of atmospheric forcings on SST predictions, motivated by their significant role in the onset of MHWs (Schlegel et al., 2021; Darmaraki et al., 2023). While this approach captures pivotal aspects, it fails to account for a variety of oceanic processes that could contribute significantly to SST variations and MHWs events. The drivers of MHWs are currently not fully understood (Holbrook et al., 2019), and the relevant physical drivers and processes involved in MHW emergence span various timescales, ranging from days (e.g., anomalous heat fluxes) to weeks (e.g., blocking systems and atmospheric teleconnections), months (e.g., re-emergence of warm anomalies from the subsurface), and years (e.g., climate modes and oceanic teleconnections). By omitting these factors, our study might provide an incomplete understanding of the intricate interplay between oceanic and atmospheric dynamics, limiting the comprehensiveness of our SST forecasting model. Introducing additional ocean variables alongside atmospheric data could enrich the predictive capacity of ML techniques, improving predictions by integrating a more comprehensive set of influencing factors.

Data-driven methods used to forecast SST and, in particular, MHW occurrence on a weekly basis, are still in their research infancy. In general, weekly MHW predictions are currently missing from the literature, although weekly forecasts of ocean conditions are widely available (Giamalaki et al., 2022). The presented work helps to demonstrate and confirm the power of these easy-to-use tools which could be efficiently applied to predict the future state of the ocean 1 week ahead. We recognize the merit of longer-term forecasting for understanding and mitigating lasting impacts, but short-term forecasts supply useful information in various sectors, specifically in industries reliant on immediate responses to temperature fluctuations (DeMott et al., 2021). Short-term SST and MHW forecasts are particularly relevant in sectors such as aquaculture, a very important activity in the Mediterranean sea, where the acute impacts of short-duration extreme events can significantly impact operations (DeMott et al., 2021; Frölicher and Laufkötter, 2018). For instance, an extreme MHW lasting a week might impose sudden stresses on temperature-sensitive aquatic species (e.g., the early life stages of fish; Buttay et al., 2023), potentially causing immediate adverse effects, such as mass mortality (Lee et al., 2018; Buttay et al., 2023; Guinaldo et al., 2023), compared

to a more prolonged, moderate MHW. Timely predictions of these short-term fluctuations enable proactive management strategies, aiding in disease prevention, optimizing feeding schedules, preventing potential coral bleaching events, and ensuring the wellbeing of farmed species and hence safeguarding economic interests and sustaining food security (Oidtmann et al., 2011; DeMott et al., 2021).

## 5   Conclusions and future work

Our study is designed as a proof of concept to showcase the potential applications of machine learning methods in short-term (up to 7 d) SST and MHWs forecasting. ML methods, especially LSTM, showed an early edge over dynamical models, marking a preliminary step toward advanced systems capturing non-linear connections. Future efforts should refine ML models by addressing data averaging limits and exploring higher-resolution data. Integrating physical laws into ML frameworks could enhance reliability, while exploring additional oceanic variables would aid in understanding dynamics, especially MHW drivers. These forecasting tools offer proactive strategies for aquaculture and ecosystem management, providing rapid warnings for stakeholders to mitigate ecosystem impacts.

*Code and data availability.*   The code generated during the current study to train and test the ML model is stored in the Zenodo repository (https://doi.org/10.5281/zenodo.8335345, Bonino et al., 2023a). All data used in this study are open access. The SST dataset used in this study is the European Space Agency (ESA) Climate Change Initiative SST dataset v2.1 (Merchant et al., 2019), and it is freely available in the CEDA catalogue at https://catalogue.ceda.ac.uk/uuid/62c0f97b1eac4e0197a674870afe1ee6 (Good et al., 2019) for September 1981 to December 2016 and in the Copernicus CDS at https://cds.climate.copernicus.eu/cdsapp#!/dataset/satellite-seasurfacetemperature?tab=overview (Copernicus Climate Change Service, 2019) for January 2017 to December 2021. The relevant atmospheric variables are taken from the European Centre for Medium-Range Weather Forecasts (ECMWF) ERA5 dataset (Hersbach et al., 2020) at https://doi.org/10.24381/cds.adbb2d47 (Hersbach et al., 2023).

*Supplement.*   The supplement related to this article is available online at: https://doi.org/10.5194/os-20-1-2024-supplement.

*Author contributions.*   GB, GG, and SM conceived the study. GB, SM, and GG discussed and defined the methodological framework. GB performed the experiment and wrote the manuscript. GB, SM, and GG interpreted the results. GB, SM, GG, RM, and EC contributed to the interpretation of the results and to the paper writing. All authors reviewed the manuscript.

*Competing interests.*   The contact author has declared that none of the authors has any competing interests.

ther geographical representation in this paper. While Copernicus Publications makes every effort to include appropriate place names, the final responsibility lies with the authors.

*Special issue statement.*   This article is part of the special issue "Special Issue for the 54th International Liège Colloquium on Machine Learning and Data Analysis in Oceanography". It is a result of the 54th International Liège Colloquium on Ocean Dynamics Machine Learning and Data Analysis in Oceanography, Liège, Belgium, 8–12 May 2023.

*Acknowledgements.*   We acknowledge the CMCC Foundation for providing computational resources.

*Financial support.*   This research has been funded by the European Space Agency (ESA) as part of the FEVERSEA Climate Change Initiative (CCI) fellowship (ESA ESRIN/contract no. 4000133282/20/I/NB).

*Review statement.*   This paper was edited by Matjaz Licer and reviewed by two anonymous referees.

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

**Remarks from the typesetter**

TS1    Please give an explanation of why this needs to be changed. We have to ask the handling editor for approval. Thanks.
TS2    Please provide date of last access.