# Peer review of "Machine learning methods to predict Sea Surface Temperature and Marine Heatwave occurrence: a case study of the Mediterranean Sea"

_EGUsphere, 2023_

## Referee Comment (RC1)

**Review Ocean Science** *Bonino et al 2023 :*
Machine learning methods to predict Sea Surface Temperature and Marine Heatwave occurrence: a case study of the Mediterranean Sea

**General comments**

In this manuscript, the authors aim to develop a short-term SST and MHW framework based on contemporary machine learning (ML) techniques, using the Mediterranean Sea as an illustrative example. This subject addresses important and urgent issues concerning the increasing number of marine heatwaves affecting this area. To achieve this goal, the authors evaluate three common ML techniques: Long Short-Term Memory (LSTM), Convolutional Neural Network (CNN), and Random Forest (Rforest).

This topic propose a method to bridge a gap in the operational prediction of MHW using a low computational data-driven approach.

In my opinion, this work is interesting and demonstrates the adaptability of these techniques to such an urgent topic. The manuscript is well-written and easy to read. However, after reading the manuscript, I find myself with more questions than answers about this study, mainly due to a lack of explanations regarding the construction of the framework and the data used. In particular, these concerns are about two topics:

First, the construction of the ML classification for extreme events. By definition, an extreme event is a phenomenon that is rare in a dataset, making it challenging to identify patterns and leading to imbalanced classes. In my opinion, this situation may explains the high number of FPR you found in the MHW prediction. This question about extreme events is crucial in the context of climate variations driven by climate change. The data used in this study seem not to be detrended, allowing extremes to originate from different sources, even when only meteorological predictors are considered. It would be beneficial to have a smaller sample in the training dataset with an acceptable bias to provide the network with more information about extremes (it a common way of doing, for example you can refer to the work by Mounier et al. 2022).

Mounier, A., Raynaud, L., Rottner, L., Plu, M., Arbogast, P., Kreitz, M., ... & Touzé, B. (2022). Detection of bow echoes in kilometer-scale forecasts using a convolutional neural network. *Artificial Intelligence for the Earth Systems*, *1*(2), e210010

The second concern relates to the study's objectives. If I read correctly, the justification of this study is the need for a prediction system for SST and MHW due to their significant social, ecological, and economic impacts: I couldn't agree more. However, I did not find any justification, and I'm not convinced by, the impact of having a short-term forecast of SST/MHW. Given the profound impact of MHWs on marine ecosystems and, consequently fisheries, and the timescales of MHWs (lasting for days to years) many researchers are focusing more on multi-weeks or seasonal forecasts to anticipate these impacts. I can speculate on why you developed this short-term solution, but it is not clearly justified in the article and seems to be useful for very specific industry. For example, does an extreme 1week MHW has more impact than a moderate MHWs that lasts 1 year. In connection with this, I would suggest exploring a comparison between these ML techniques and multiple linear regression using the same predictors. This would help justify choosing ML over a statistical model with lower computational costs, as you mentioned in Page 2, Line 42.

All of these points highlight the need for a more comprehensive discussion and justification of this study, which undoubtedly has some very interesting findings. Despite the length of this review, I'm convinced by the importance of such work and the publication in Ocean Science.

**Line by line comments**

**Abstract**

L13:
Regarding your results, you could not conclusively state that ML techniques 'outperform' MedFS, even in the context of 3-day predictions. In my opinion, the results fall within the margin of error (please refer to my comments in the Results section). Furthermore, you should replace 'most regions' with a quantitative assessment of this outcome.

I also suggest adding a closing sentence to your abstract to provide some perspective. However, the decision is up to you.

**Introduction**

As mentioned in the general comments, the introduction lacks justification for considering only a 7-day forecast period. If we compare this to other scientific fields, such as numerical weather prediction, the justification for using ML techniques often leans in the opposite direction: attempting to outperform models beyond the 7-day threshold, where predictability reaches its limits. In my opinion, this approach would be even more compelling for SSTs, given the relatively small range of variation within a week, especially in your case when utilizing interpolated L4 data for training.

**P1.L18-21**

I would recommend rewriting the introduction to separate the sections addressing SST, the impact on ecosystems, and the definition of MHWs as extreme events. In my opinion, there is a bit of confusion (at least as a reader) between these concepts in the initial part of the introduction, and you are defining them a second times in the Results section (Page 8, Line 229).

**P2.L39**
I'm not familiar with the reference you mentioned but I have in mind one counter example. In numerical weather prediction AI models like FourCastNest, the model provides forecasts globally for up to 15 days using ML techniques exclusively. This raises an important questions about the applicability of your study, especially given the wide area you are examining. Following this sentence it would be preferable to use numerical approach in the case of Med Sea and I could not find the justification in your introduction.

**P2.L44-65**

I think you could shorten this section which is currently a very long review. As you mentioned, statistical techniques have a well established history in SST including bias estimation and satellite data reconstruction.

Saux Picart, S.; Tandeo, P.; Autret, E.; Gausset, B. Exploring Machine Learning to Correct Satellite-Derived Sea Surface Temperatures. Remote Sens. 2018, 10, 224. https://doi.org/10.3390/rs10020224

Barth, A., Alvera-Azcárate, A., Licer, M., and Beckers, J.-M.: DINCAE 1.0: a convolutional neural network with error estimates to reconstruct sea surface temperature satellite observations, Geosci. Model Dev., 13, 1609–1622, https://doi.org/10.5194/gmd-13-1609-2020, 2020

**Methodological Framework**

- One of my main concerns in this section deals with the choice of the data and its inherent justification. If I understand correctly, the aim of the study is to develop a SST forecast up to 7 days. However, L4 ESA CCI SST product is primarly designed for climate studies and is a gap-free product, meaning that some of the data are computed using interpolation. By using such data, you may inadvertently introduce biases into your model. For example, data are smoothed and you may hide upwelling in coastal areas.
  An alternative approach could have been to use dedicated L3 operational data (near real time product), such as those developed by the same ESA CCI project or the OSI SAF project.

- In connection with this, how do you account for the differences in spatial resolution between the ESA CCI and ERA5 datasets? You are attempting to link SST variations to atmospheric predictors but ERA5 has a spatial resolution of 0.25° whereas ESA CCI provides SST products at a resolution of 0.05°.
  I might have missed something in the study but it appears to me that you are calculating a mean SST for each area. Do you have a single SST representative for each zone?
  I have the same question with the atmospheric variables. However these zones represent large areas and by aggregating data in this way you may smooth out the data. Thereby by neglecting the non-linearity of wind speed, which is a key driver of SST variations especially in the Mediterranean Sea where you have regional winds that interact strongly with SSTs.

- I'm also surprised that you did not account for 2m air surface temperature, specific humidity and mixed layer depth variables as they are known drivers of MHWs and even more in the Mediterranean Sea. You took T2m and q2m into account in the sensible/latent heat fluxes but it would be interesting to know about their direct impact. I'm also referring to the MLD because MHWs often have a vertical extension and can persist in intermediate layers without necessarily having any signature at the surface. See:

Amaya, D. J., Miller, A. J., Xie, S.-P., and Kosaka, Y.: Physical drivers of the summer 2019 North Pacific marine heatwave, Nat. Commun., 11, 1903, https://doi.org/10.1038/s41467-020-15820-w, 2020.

Chen, HH., Wang, Y., Xiu, P. *et al.* Combined oceanic and atmospheric forcing of the 2013/14 marine heatwave in the northeast Pacific. *npj Clim Atmos Sci* **6**, 3 (2023). https://doi.org/10.1038/s41612-023-00327-0

Guinaldo, T., Voldoire, A., Waldman, R., Saux Picart, S., and Roquet, H.: Response of the sea surface temperature to heatwaves during the France 2022 meteorological summer, Ocean Sci., 19, 629–647, https://doi.org/10.5194/os-19-629-2023, 2023

Eric C.J. Oliver, Jessica A. Benthuysen, Sofia Darmaraki, Markus G. Donat, Alistair J. Hobday, Neil J. Holbrook, Robert W. Schlegel, Alex Sen Gupta, Annual Review of Marine Science 2021 13:1, 313-342

- One of the advantages of the numerical approach is its ability to account for a wide variety of processes that are parameterized within the model. However, here you are only exclusively studying SST predictions within the scope of the atmospheric forcings. How do you account for oceanic processes? Given their potential significance? For example, smoothing wind speed variations might reduce the contribution of the vertical mixing among other crucial oceanic processes. You mention about this in your discussion but this could be extended.

- Regarding the detection of MHWs detection, the methodology is not entirely clear for me. Firstly, I could not find any explanation of how you calculate the daily climatology and the seasonal threshold. Are these values unique for each regions or are they calculated for every pixels at the ESA CCI resolution? Do you consider an MHW to occur when at least 1 pixels within a region exceeds the threshold or do you consider the mean SST over the region?
Furthermore, calculating a climatology by only taking the mean of daily SSTs over 30 years can introduce some biases because it may not be statistically robust and sensitive to extremes. Typically the climatology is computed for a specific day using 30 years of data either with an 11-day sliding window centered on the day in question or by employing the first harmonic of a Fourier series.
Do you account for situations when a MHWs occurs less than 2 days before the end of another one it is considered part of the MHW?
**P6.L175:** you mention computing the climatology over the period 1981-2016 which is not the international standard of using 30 years. It is worthy noting that the first complete year of ESA CCI data is 1982. More generally, you should clarify how you calculate the climatology and anomalies.

- In studying MHWs, it is important not only to detect them but also to estimate key metrics such as the mean intensity, max intensity and the severity. Do you have insights into the possibility of predicting these metrics using your model?

**P4.L104**

This part should be moved to the results and discussions sections. Moreover, GEO and INC are not independent variables, which may explains why they are correlated with SST. Even though you are looking at daily mean, ESA CCI SST are representative of the foundation temperature which is not influenced by the diurnal cycle. Therefore, you should rephrase the sentence 'INC influenced directly SST during daily time' (P4.L7). I understand what you meant and suggest this modification to avoid the reader to misunderstood.

I'm not entirely convinced that MI calculations are necessary to identify relations between variables. Examining the upper-ocean layer equation could give the same result (although it is not quantitative). Additionally, it is excepted to find a weak correlation between wind speed and SST due to the non-linearity I mentioned earlier which you did not take into account.

**P5.L124-141**

I have several questions regarding the description of the networks you used.

Firstly, you have not detailed the inputs/ouputs and how the variables are incorporated in the networks (possibly as concatenated large vectors?).
Additionally, I found no indications in this work regarding how the data were normalized to make different atmospheric and oceanic variables comparable.
Additionally, I find it somewhat surprising that you used 200 epochs for both LSTM and CNN without discussing the complexity of the networks. For instance, in atmospheric models, LSTM converge quickly compared to CNN and it is common practice to add an early stopping to prevent overfitting.

**P6.L158**
This is a minor comment but you mentioned computational time without providing hardware specifications. I may find this information useful.

**Equation 4:** There is a typo in the F1 score, it is 2*P*R/(P+R)

Results

- I'm a bit surprised by the lack of stability in the ML techniques (this might be related to a lack of experience with these techniques from my side). On the contrary, as you mentioned in P7.L210, it is normal for the dynamical model to be rather stable for up to 7 days (in my opinion it may depends on the predictability limits). Are you sure that the behavior of your ML models is not influenced by some sort of persistence or memory effect that could explain the increase in RMSE? I thought about this because you may have introduced some biases during the training stage, as I mentioned earlier. Additionally, you are also studying SST through a spatial average over large areas. Furthermore, at the first order, a 0.1°C difference is within the common measurement errors in satellite SST products.
  Regarding all this, I would not claim that ML techniques 'outperformed' the numerical model, but as you mentioned in P8.L213, they 'compare favorably'.

- In a second step, your study and the results are quite interesting. In my opinion, it could be improved by an in-depth analysis of the results by regions, attempting to understand the diversity of responses from different basins (explaining it through dynamic conditions or other factors). For example, the dynamic in regions like the Alboran Sea may explain most of the variability. I understand that it may not be feasible to include everything in a single study, but this could be explored in the context of an additional study.

- **Figure 5** is very informative, would it be possible for you to add the daily climatological mean in addition to the threshold. It is just a suggestion but it would be also interesting to add a focus on a particular MHWs because it seems that FigS3 shows some sort of time lag in the SST between models. Maybe add some metrics such as the correlation, annual mean and standard deviation.

- **Figure 6:** I found it a bit difficult to read the figure with two distinct labels for SST and the other variables. In addition, the lines are very thin, and I can't distinguish easily between the variables.

**Discussion/Conclusion**

In general, the discussion is good, with some very interesting points and references. I appreciate that you are not attempting to oversell your results. ML techniques, despite some limitations, have demonstrated predictive skill, and your discussion about the challenges in understanding MHW, along with a thoughtful exploration of time-scales, raised some intriguing questions.

**L293-295:** In regard of your results the ML techniques seems to have better result however regarding the stability, I'm not convinced that it is not linked to some memory effect. I'm also not sure of the impact of predicting SSTs at 1 and 3 days, except for very specific industry with high tolerance to the risks.

**L310**: I agree with the contribution of incoming solar radiation, which is generally linked to lower than average anomalies in cloud cover. However, this contribution is limited in summer in the Mediterranean Sea due to the usual low cloud cover over the area. Thus, MHWs are primarily driven by other variables, such as heat fluxes (namely the atmospheric variables T2M, Q2M and WS) which exhibit significant regional dependencies (Guinaldo et al., 2023, as mentioned earlier). Additionally, you could have also discussed the possibility of incorporating ocean heat content to improve the forecast (Holbrook et al., 2020).

Another important point to consider is related to the data you used. As mentioned earlier, you employed SST data dedicated to climate studies instead of near-real-time data. Your discussion would benefit from addressing the limitations associated with using such data. In the context of the study framework, it would also be worthy to discuss the comparison between this type of forecasting model and other approaches, such as multiple linear regression or model ensembles.

To enhance this discussion, I recommend reading the following study:

Benthuysen, J. A., Smith, G. A., Spillman, C. M., & Steinberg, C. R. (2021). Subseasonal prediction of the 2020 Great Barrier Reef and Coral Sea marine heatwave. *Environmental Research Letters, 16*(12), 124050.

---

## Referee Comment (RC2)

Title: Machine learning methods to predict Sea Surface Temperature and Marine Heatwave occurrence: a case study of the Mediterranean Sea

Authors: Bonino et al., 2023

Scientific significance:
The manuscript describes a set of experiments obtained by training shallow (Random Forest, RForest) and deep learning (Long short-term memory, LSTM and Convolutional Neural Network,CNN) methods using ESA CCI SST and ECMWF datasets. The main goal is to predict Sea Surface Temperature and Marine Heat Waves in the Mediterranean Sea. The duration of the prediction is 7 days. Results are provided by using daily RMSE values.

Scientific quality:
The scientific question raised in this study is clear (Line 6 "*create competitive predictive tools"* and Line 69 *provide a proof-of-concept study on the advantage of data-driven ML methods*).
The work is relevant for the novelty of the aim, of the methods and for its reversibility in being used in different scientific fields.

With the advancement of NN algorithms, it is important to verify whether and to what extent data-driven methods can provide comparable products with respect to classical numerical methods.
Although the scientific contribution of this study is significant, some scientific choices need further explanations. In my opinion, the manuscript (especially the introduction) lacks useful information. This could diminish the importance of the work, especially of the methods, which I found particularly robust.

Presentation quality:
The manuscript is generally well presented and well written. Since the scientific value of this study is high, my suggestion is to add information that will enrich your study and efforts. In the *General comments* below, I have tried to list them. I hope it will be useful.

General comments:
I believe that some scientific choices need further explanations.

1) I suggest answering the following questions by adding the information in the INTRODUCTION

Both NNs and numerical models are able to cover 7 days of forecast. While the NNs take less computational time, the numerical models have higher accuracy in the last days of the forecast (SST).

- What are the main advantages of using NN to predict ocean physics? Are the advantages only computational ones?  (NN can approximate nonlinear function (Hornik et al., 1989)

- What are the main limitations in applying NN to real-world scenarios? (e.g. NN algorithms do not know when they are violating the laws of physics, e.g. Buizza et al., 2022)

The goal of the work is to "*create competitive predictive tools*". You use the RMSE to assess the feasibility of NNs in predicting SST and MHWs and you validate all the NN tests (also) comparing the results with a numerical model.

- In this case I believe that the Figures (time-series) you have chosen are sometimes contradictory and sometimes do not show what you are describing (see detailed comments)

Some interesting information on this work can be found in the discussion. I expected to read them in the introduction (see detailed comments).

2) I suggest answering the following questions by adding the information directly in the METHODS section:

In the text I have the impression that you are sometimes contradicting your scientific choices. For example when you write: "*For instance, the numerical approach is better suited for predictions over a wider area, while the data-driven techniques are more applicable for location-specific studies*". Based on this:

- Why are you proposing predictions *over a wider area with data-driven techniques*?

Since the MedFS forecast used for validation covers 10 days:

- Please explain why you chose a time interval of 7 days for the forecast (instead of 10 days).
- Have you considered or tried to extend the duration of the prediction?
  Following your results, I can infer that by extending the duration of the experiment (10days) for SST:
  NN's RMSE >> MedFS RMSE
  NN's computational time << MedFS computational time
  (please specify the computational time of the model in the text).

3) I suggest answering the following questions by adding the information directly in the DISCUSSION section:

I have the impression that the reader should follow the NN "branch" instead of the numerical models (to predict SST and MHWs). But "NN algorithms do not know when they are violating the laws of physics". On the basis of this:

- Do you think that this limits the NNs-skills along the prediction interval?
- Have you considered adding physical constraints to your NN models?
- Have you considered the possibility of predicting SST and MHW by modifying the inputs of the NNs? (e.g. by adding ocean variables to atmospheric variables)

- Would you suggest using NNs to predict SST and MHWs in a 3/5-days interval and MedFS for a longer period?

Buizza, Caterina, et al. "Data learning: Integrating data assimilation and machine learning." *Journal of Computational Science* 58 (2022): 101525.

Detailed comments (minor):

Abstract:

To facilitate the reader, you might explicitly write the number of the Experiments.

- L 4:  MHWs acronym already introduced in Line 1
- L 9: Following the outline of the manuscript I think it could be clearer to insert in L9 the sentence L 15-16
- L 11 typo error Cat °C at
- L 11 and CNN RMSE?

Introduction

I appreciated the description of the impact of MHWs on ecosystems, but since the article does not deal with ecological studies I would expect to have: (i) less information on the ecological impacts (1-2 lines) (ii) more information on the techniques chosen and their pros and cons, including a comparison with numerical models (iii) a brief overview of the main characteristics of the Mediterranean Sea with respect to the objective of your work (e.g The reason why you have chosen the Mediterranean Sea as your study area?). Please describe if there is a sector (western-central-eastern) that is more susceptible to these events.

About 25% of your abstract focuses on NN's improvements compared to numerical models (MedFS) which you mention in P9. I suggest adding a sentence on the MedFS model used to compare the results (at line 70).

- L 18-29: I think this part could be shortened.
- L 23: Garrabou et al., 2022   double citation or missing bibliography ?
- L 33: typos error  *generations(Leroux*
- L 39: Explain your motivation with respect to your work: "..*data-driven techniques are more applicable for location-specific studies.".*
- L 40-41: I suggest removing the sentence because it seems to be out of topic
- L 55  typos error *begging*
- L 44-65: If in your opinion it is important, please highlight if there is a method (shallow/deep) that better fits the SST/MHWs prediction goals (from literature).
- L 44-65: Too many citations. I suggest reducing the number of citations and adding more detailed information (i.e. the duration of the predictions from the cited literature). Moreover, in many works you cited (Corchado ,Liu, Xie etc.) the duration of the prediction is higher than 7 days.

Methodological Framework:
Data collection and preprocessing

- L 83: Add the spatial resolution of the datasets e.g. "daily satellite-derived Sea Surface Temperature (SST) data".
- L 86: the L4 dataset provides interpolated data (L3 does not). I assume that you have decided to use the L4 dataset to numerically enrich the dataset for training and test evaluation. Is this the case?
- L 102-114: In my opinion, table 1 is a result (as you write in the abstract L15-16)

Experiments/Evaluation metrics :

- L 147: I would have introduced MedFS earlier in the text (e.g. in the Introduction)
- L 173: remove citation in parentheses "*in Hobday et al. 2016 (Hobday et al., 2016)*"
- L. 182: you refer to a table where I guess there's a typo in the first column-second row cell: *MH predicted*

Results:
If you are going to redo some figures, please increase the font size (especially for the legend).

- L 198: "For instance, in all the techniques, region 15 shows the lowest, or almost the lowest (e.g in CNN) RMSE, while region 11 shows the highest errors"
  I cannot recognise region 15 in the CNN subplot, it is hardly readable. From figure 3 I can only evaluate the daily variability of RMSE across the Med. basins. Perhaps consider that plotting the average RMSE would be more useful to have a general overview on which method 'outperforms' the others.
- L. 198-200: Is there a reason why RMSE in region 11 is higher and 15 lower? A brief overview of the main physical characteristics of the Mediterranean Sea in the introduction might be helpful.
- L. 205: "*They also represent different dynamical areas of the Mediterranean basin.*" Which ones? Add information in the Introduction.
- L. 210: "*In contrast to ML methods, the dynamical model's prediction of SST is influenced by atmospheric forecasts throughout the forecast period, which likely prevents the RMSE from increasing with the lead time.*"
  I think this is not a result, I would rather move it into the Discussion section.
- L 215: "*ML methods show lower RMSE than the MedFS forecast system for the first 3 days of forecast and they are comparable at lead time of 5 days.*"
  From Figure 4a
  - it results that the RMSE of MedFS at day3 is less than the one from NN methods in west and east only.
  - why are MedFS bullets plotted only for days 1,3,5?
  - UEXPs are called whiskers in the Figure and bar / UEXPs in the text. Better to be consistent with the nomenclature.
- L 221: "*It is likely connected to the fact that CNN algorithms are typically designed for image processing rather than time-series forecasting*". It would probably be better to have this information right from the introduction.

- L 223: Is there a particular reason to choose the year 2020?
- L 225: *The figure shows a very close match between the forecasts and the observations.* Hence NNs don't outperform the MedFS.
- L 232: "*while Figure 4b shows the variation of the F1 score for all the methods with increasing forecast lead time*"
  The reader expects at this point an explanation of the figure 4b. It comes at L244. I suggest removing this sentence or adding information previously.
- Figure 6: impossible to see the lines. Since you only explain INC SST and LAT why plotting the others atmo forcing? (you can merge all the other ones in 1 single line)
- L264 and L267: add abbreviation in the text (INC in fig… lat in.. )

Discussion and conclusions:

Can you introduce possible future developments?

- L278-282: All the abbreviations are already introduced
- L299-301: These features are typical of all the NNs methods. I would prefer to have this general info in the Introduction section.
- L330-336: Too much information on the impact, I suggest reducing this paragraph.

Supplementary:

Correct the description of Figure S4
*Figure S4: As Figure S4 but for forecast lead time 3.*

---

## Author Comment (AC1)

Dear Referee,

we would like to thank you for the careful reading of the manuscript and the constructive comments that substantially helped to improve and clarify the paper. Answers to all your comments are detailed hereafter. Corrections to the English grammar were adopted in the revised version of the manuscript according to the reviewer's recommendations, but are not reported or discussed here. All authors agree with the modifications made to the manuscript. The comments by the referee are reported in bold followed by our response (in blue). The text added to the revised manuscript is reported in italic font. The line numbers reported in the answers referred to the revised manuscript. The revised manuscript that includes track changes is also provided in pdf format.

In the following answers, we use 'Figure' to identify the figures in the updated manuscript and we use 'Plot' to identify the figures in this document.

**General comments:**
**I believe that some scientific choices need further explanations.**
**1) I suggest answering the following questions by adding the information in the INTRODUCTION**
**-Both NNs and numerical models are able to cover 7 days of forecast. While the NNs take less computational time, the numerical models have higher accuracy in the last days of the forecast (SST).**
**- What are the main advantages of using NN to predict ocean physics? Are the advantages only computational ones? (NN can approximate nonlinear function (Hornik et al., 1989)**
**- What are the main limitations in applying NN to real-world scenarios? (e.g. NN algorithms do not know when they are violating the laws of physics, e.g. Buizza et al., 2022)**
**The goal of the work is to "create competitive predictive tools". You use the RMSE to assess the feasibility of NNs in predicting SST and MHWs and you validate all the NN tests (also) comparing the results with a numerical model.**
**- In this case I believe that the Figures (time-series) you have chosen are sometimes contradictory and sometimes do not show what you are describing (see detailed comments)**
**Some interesting information on this work can be found in the discussion. I expected to read them in the introduction (see detailed comments).**

Thank you for your thorough review and insightful comments regarding our study's objectives and introduction. We extended the introduction taking into account your comments. We extend the text at line 33:"...*In recent years, increasing interest has been given to machine learning (ML) techniques, even though, in contrast to the dynamical model, ML techniques do not know when they are violating the laws of physics (Buizza et al., 2022). As a "learning from data" approach, machine learning has the advantages of computational efficiency, accuracy, transferability, flexibility and ease-of-use in ocean forecasting studies (Boukabara et al., 2019; Li et al., 2020; Wei and Guan, 2022; Taylor and Feng, 2022), Moreover, it is also less prone to model bias errors (Jacox et al., 2020) and, beyond computational efficiency, ML techniques excel in approximating nonlinear functions (Hornik et al. 1989).* "

**2) I suggest answering the following questions by adding the information directly in the METHODS section:**
In the text I have the impression that you are sometimes contradicting your scientific choices. For example when you write: "For instance, the numerical approach is better suited for predictions over a wider area, while the data-driven techniques are more applicable for location-specific studies" (L.39). Based on this:
- Why are you proposing predictions over a wider area with data-driven techniques?
Since the MedFS forecast used for validation covers 10 days:
- Please explain why you chose a time interval of 7 days for the forecast (instead of 10 days).
- Have you considered or tried to extend the duration of the prediction? Following your results, I can infer that by extending the duration of the experiment (10days) for SST:
NN's RMSE >> MedFS RMSE
NN's computational time << MedFS computational time
(please specify the computational time of the model in the text).

Thank you for your suggestions.For the sentence you are referring to, there was a misinterpretation in our statement. We initially perceived "location-specific" studies as pertaining to time-series analyses. Consequently, we've removed this sentence from the introduction to avoid confusion. We extended the methodology taking into account your comments. We extend the text at line 146: *"MedFS predictions, part of the Copernicus Marine Service since 2017, offer SST forecasts for lead times up to 9 days averaged across specified regions. Since the accuracy of the ML forecast decreases almost linearly with time, we decided to limit the comparison with the MedFS forecasted SST averages for lead times of 1, 3, and 5 days. "*

We want to highlight to the reviewer that the MedFS forecasts are operationally provided as averaged SST time-series over the selected regions for lead times of 1,3,5 and 9 days.

The computational time for MedFS is 10 minutes to simulate 1 day. We added this information in the manuscript at line 163.

**3) I suggest answering the following questions by adding the information directly in the DISCUSSION section:**
I have the impression that the reader should follow the NN "branch" instead of the numerical models (to predict SST and MHWs). But "NN algorithms do not know when they are violating the laws of physics" . On the basis of this:
- Do you think that this limits the NNs-skills along the prediction interval?
- Have you considered adding physical constraints to your NN models?
- Have you considered the possibility of predicting SST and MHW by modifying the inputs of the NNs? (e.g. by adding ocean variables to atmospheric variables)
- Would you suggest using NNs to predict SST and MHWs in a 3/5-days interval and MedFS for a longer period?

**Buizza, Caterina, et al. "Data learning: Integrating data assimilation and machine learning." Journal of Computational Science 58 (2022): 101525.**

We extended the discussion taking into account your comments. We added some text at line 347: "*Furthermore, the neural network algorithms applied lack inherent knowledge of physical laws, potentially leading to violations of fundamental physical constraints. This limitation likely restricts the proficiency of NNs in maintaining accuracy throughout the prediction interval (Boukabara et al. 2019, Dueben et al. 2018). It prompts a need for cautious interpretation and validation of results, especially in scenarios where physical constraints significantly influence outcomes. While this study primarily focuses on predictive skill, future iterations could explore methodologies to infuse ML frameworks with physical laws governing ocean dynamics. For instance, Zanetta et al. (2023) propose achieving physical consistency in deep learning-based postprocessing models for temperature and humidity by incorporating meteorological expertise through analytic equations. Incorporating these constraints could potentially enhance the reliability of NN predictions, mitigating the risk of straying from physical realities. One may consider, due to the strengths of NNs in shorter-term predictions (3/5-days interval) and the potential limitations discussed, to use the ML model alongside models like MedFS for longer prediction intervals.. Utilizing MedFS for longer-term forecasts could leverage its established reliability over extended periods, while ML could excel in shorter-term predictions.*"

**Detailed comments (minor):**
**4) Abstract:**
**To facilitate the reader, you might explicitly write the number of the Experiments.**
**● L 4: MHWs acronym already introduced in Line 1**
**● L 9: Following the outline of the manuscript I think it could be clearer to insert in L9 the sentence L 15-16**
**● L 11 and CNN RMSE?**

Thank you for your comments, we rephrased the abstract:

"*Marine heatwaves (MHWs) have significant social and ecological impacts, necessitating the prediction of these extreme events to prevent and mitigate their negative consequences and provide valuable information to decision-makers about MHW-related risks. In this study, machine learning (ML) techniques are applied to predict Sea Surface Temperature (SST) time series and Marine Heatwaves in 16 regions of the Mediterranean Sea. ML algorithms, including Random Forest (RForest), Long short-term memory (LSTM), and Convolutional Neural Network (CNN), are used to create competitive predictive tools for SST. The ML models are designed to forecast SST and MHWs up to 7 days ahead. For each area, we performed 15 different experiments for ML techniques, modifying the training and testing periods. Alongside SST, other relevant atmospheric variables are utilized as potential predictors of MHWs. Datasets from the European Space Agency Climate Change Initiative (ESA CCI SST) v2.1 and the European Centre for Medium-Range Weather Forecasts (ECMWF) ERA5 reanalysis from 1981 to 2021 are used to train and test the ML techniques. For each area the results show that all the ML methods performed with minimum Root Mean Square Errors (RMSE) of about 0.1°C at a 1-day lead time and maximum values of about 0.8°C at a 7-day lead time. In all regions, both the RForest*

*and LSTM models consistently outperformed the CNN model across all lead times. LSTM has the highest predictive skill in 11 regions at all lead times. Importantly, the ML techniques show results similar to the dynamical Copernicus Mediterranean Forecasting System (MedFS) for both SST and MHW forecasts, especially in the early forecast days. For MHW forecasting, ML methods compare favorably with MedFS up to 3-day lead time in 14 regions, while MedFS shows superior skill at 5-day lead time in 9 out of 16 regions. All methods predict the occurrence of MHWs with a confidence level greater than 50\% in each region. Additionally, the study highlights the importance of incoming solar radiation as a significant predictor of SST variability along with SST itself.*"

**5) Introduction**
**I appreciated the description of the impact of MHWs on ecosystems, but since the article does not deal with ecological studies I would expect to have: (i) less information on the ecological impacts (1-2 lines) (ii) more information on the techniques chosen and their pros and cons, including a comparison with numerical models (iii) a brief overview of the main characteristics of the Mediterranean Sea with respect to the objective of your work (e.g The reason why you have chosen the Mediterranean Sea as your study area?). Please describe if there is a sector (western-central-eastern) that is more susceptible to these events.**

Thank you for your feedback, we considered all your comments. We choose to retain the impacts paragraph as we believe this information is crucial for comprehending the objectives of our work, following reviewer1's observation. We add specifications on ML methods at line 50: " … *It is important to note that within the methods explored in this paper, LSTM stands out as the sole technique explicitly crafted for handling time series data. In contrast, CNN algorithms are commonly tailored for image processing tasks and are not inherently designed for time-series forecasting. To the best of our knowledge, just a few studies employed the RForest model to predict sea surface temperature…*" . We add justification of the choice of Mediterranean Sea at line 70: "*The Mediterranean Sea is a well-studied hot spot for MHW events (Garrabou et al., 2009; Giorgi, 2006; Cramer et al., 2018; Pastor et al., 2020; Pastor and Khodayar, 2022; Garrabou et al., 2022; Ciappa, 2022). As detailed in the study by Bonino et al. (2023), the Adriatic Sea, the Gulf of Lion, and the Alboran Sea encountered the highest occurrence, shortest duration, and most severe MHWs. The Mediterranean Sea serves our study area due to its relevance for marine management activities, For instance, over 95% of the global production of seabream and seabass comes from aquaculture, with Mediterranean countries contributing to 97% of this production. (Carvalho, N., & Guillen, J. 2021).* "

**About 25% of your abstract focuses on NN's improvements compared to numerical models (MedFS) which you mention in P9. I suggest adding a sentence on the MedFS model used to compare the results (at line 70).**

**● L 18-29: I think this part could be shortened.**
**● L 44-65: If in your opinion it is important, please highlight if there is a method (shallow/deep) that better fits the SST/MHWs prediction goals (from literature).**
**● L 44-65: Too many citations. I suggest reducing the number of citations and adding more detailed information (i.e. the duration of the predictions from the**

**cited literature). Moreover, in many works you cited (Corchado ,Liu, Xie etc.) the duration of the prediction is higher than 7 days.**

Thanks for your feedback. We've considered all your comments and, indeed, made substantial revisions to the literature review. We rephrased at line 46: "... *Pioneering works on SST prediction using deep learning methods are, instead, more recent. In the last few years, as reported in a recent review paper (Haghbin et al., 2021), the deep learning-based models such as the RNN, the Long Short Term Memory (LSTM, e.g. Xiao et al. 2019, Liu et al. 2018 and Xie et al. 2019), and CNN (e.g. Han et al. 2019) have attracted progressively more attention in the research community, providing accurate estimates among the models considered…*"

**6) Methodological Framework:**
**Data collection and preprocessing**
**● L 83: Add the spatial resolution of the datasets e.g. "daily satellite-derived Sea Surface Temperature (SST) data".**
**● L 86: the L4 dataset provides interpolated data (L3 does not). I assume that you have decided to use the L4 dataset to numerically enrich the dataset for training and test evaluation. Is this the case?**
**● L 102-114: In my opinion, table 1 is a result (as you write in the abstract L15-16)**

Thank you for your comments. We added the resolution of the ESA CCI SST dataset at line 86: "*This dataset consists of daily maps of average SST at 20 cm nominal depth with 0.05 × 0.05 of horizontal resolution, covering the period from September 1981 to December 2016.*". We added the motivation for the L4 dataset at line 304: "*In this study, a group of ML algorithms - Random Forest (RForest), Long short-term memory (LSTM) and Convolutional Neural Networks (CNN) - are used to evaluate their ability in building a competitive prediction tool of SST and MHWS occurrence 7 days ahead in the Mediterranean Sea. The methods use the European Space Agency (ESA) Climate Change Initiative (CCI) Sea Surface Temperature, Sea Level Pressure (SLP), Geopotential Height at 500hPa (GEO), Wind Speed (WS), Sensible Heat flux (SENS), Latent Heat flux (LAT) and incoming solar radiation (INC) from ECMWF ERA5 as input data. We compared the ML predictions against the MedFS prediction system, part of the Copernicus Marine Service since 2017, which offers SST forecasts for lead times up to 9 days averaged across specified regions. It is important to underline that the data used in our work are designed primarily for climate studies and providing a gap-free dataset through interpolation, raises concerns about potential biases introduced into our forecasting model. The interpolation process, while ensuring a comprehensive dataset, might inadvertently smooth variations and obscure critical phenomena like coastal upwelling. Alternative approaches utilizing near real-time operational data, could offer more dynamically responsive and less biased datasets for improved forecasting accuracy. Moreover, our methodology involves averaging SST data to obtain a single representative SST for each zone and the same approach is used for atmospheric predictors. We acknowledge that averaging data in this manner might smooth out localized variations, thereby potentially overlooking the non-linearities of*

*variables such as wind speed*". We put Table 1 in the results section titled "Mutual information analysis".

**7) Experiments/Evaluation metrics :**
**● L 147: I would have introduced MedFS earlier in the text (e.g. in the Introduction)**
We also added a sentence to introduce the MedFS in the introduction at line 67: "*The ML methods are compared against the Copernicus Mediterranean Forecasting System (MedFS, i.e. dynamical ocean model, (Clementi et al., 2021). MedFS is a numerical ocean prediction system, implemented and developed by the Euro-Mediterranean Center on Climate Change (CMCC), that produces analyses and short term forecasts for the entire Mediterranean Sea and adjacent areas in the Atlantic Ocean (Clementi et al., 2021).* "

**8) Results:**
**If you are going to redo some figures, please increase the font size (especially for the legend).**
**● L 198: "For instance, in all the techniques, region 15 shows the lowest, or almost the lowest (e.g in CNN) RMSE, while region 11 shows the highest errors" I cannot recognise region 15 in the CNN subplot, it is hardly readable. From figure 3 I can only evaluate the daily variability of RMSE across the Med. basins. Perhaps consider that plotting the average RMSE would be more useful to have a general overview on which method 'outperforms' the others.**
**● L. 198-200: Is there a reason why RMSE in region 11 is higher and 15 lower? A brief overview of the main physical characteristics of the Mediterranean Sea in the introduction might be helpful.**
**● L. 205: "They also represent different dynamical areas of the Mediterranean basin." Which ones? Add information in the Introduction.**

Thank you for your input. We decided to retain Figure 3 but we added Figure 1 as Figure 4 in the manuscript. Following your suggestion, we decided to plot the mean RMSE error for the LSTM REXPs experiment for each region (Figure 1). We can notice that the areas with larger errors are: (i) the Alboran Sea strongly, characterized by a complex dynamics influenced by the incoming cold Atlantic water through the Gibraltar Strait modulating the water transport; (ii) the North-West part of the basin, which is an area of dense water formation and intense dynamics due to the Gulf of Lion gyre (Madec et al., 1991; Pinardi et al., 2006) and by a boundary intense current called the Liguro-Provenal-Catalan Current (Pinardi et al., 2006); (iii) the Adriatic Sea, especially in its northern shelf, characterized by a complex topography, intense air-sea exchanges, large riverine inputs that contribute to enrich the dynamics of the area. Thus the correct SST representation and forecast in these areas represent a challenging modeling problem. We added Figure 1 in the manuscript and reported this description at line 219. We also added the RForest and CNN mean error as Figure S2.

[Figure]

Figure 1. LSTM mean RMSE for REXP for each region

● **L. 210: "In contrast to ML methods, the dynamical model's prediction of SST is influenced by atmospheric forecasts throughout the forecast period, which likely prevents the RMSE from increasing with the lead time."**
**I think this is not a result, I would rather move it into the Discussion section.**
● **L 215: "ML methods show lower RMSE than the MedFS forecast system for the first 3 days of forecast and they are comparable at lead time of 5 days."**
**From Figure 4a**
**- it results that the RMSE of MedFS at day3 is less than the one from NN methods in west and east only.**
**- why are MedFS bullets plotted only for days 1,3,5?**
**- UEXPs are called whiskers in the Figure and bar / UEXPs in the text. Better to be consistent with the nomenclature.**
● **L 221: "It is likely connected to the fact that CNN algorithms are typically designed for image processing rather than time-series forecasting". It would probably be better to have this information right from the introduction.**
● **L 223: Is there a particular reason to choose the year 2020?**
● **L 225: The figure shows a very close match between the forecasts and the observations. Hence NNs don't outperform the MedFS.**
● **L 232: "while Figure 4b shows the variation of the F1 score for all the methods with increasing forecast lead time"**
**The reader expects at this point an explanation of the figure 4b. It comes at L244.**
**I suggest removing this sentence or adding information previously.**

Appreciate your feedback. We've considered your comments while revising the results and discussion section. The inclusion of the 2020 year doesn't stem from a specific motivation but rather its availability from MedFS and the forecasted time-series. In Figure 4a, we've plotted bullets solely for days 1, 3, and 5 as these correspond to the available lead times in MedFS, as previously explained in the methodology section at line 147: "*MedFS predictions, part of the Copernicus Marine Service since 2017, offer SST forecasts for lead times up to 9 days averaged across specified regions. Since the accuracy of the ML forecast decreases almost linearly with time, we decided to limit the comparison with the MedFS forecasted SST averages for lead times of 1, 3, and 5 days.*"

As said before, we also added a sentence to introduce the MedFS in the introduction at line 66: "*The ML methods are compared against the Copernicus Mediterranean Forecasting System (MedFS, i.e. dynamical ocean model, (Clementi et al., 2021).*

*MedFS is a numerical ocean prediction system, implemented and developed by the Euro-Mediterranean Center on Climate Change (CMCC), that produces analyses and short term forecasts for the entire Mediterranean Sea and adjacent areas in the Atlantic Ocean (Clementi et al., 2021).* "

● **Figure 6: impossible to see the lines. Since you only explain INC SST and LAT why plotting the others atmo forcing? (you can merge all the other ones in 1 single line)**

Thank you, we simplified  the Figure.  The Figure 2 below is now the Figure 7 in the manuscript.  We explained it at line 287:
"*The labels of Figure 6 indicate, for each experiment, the driver that has been shuffled in REXPs. For an easier interpretation of the results we are showing only the SST, the INC and the LAT drivers, as they are the most relevant.* ". We comment on this Figure at line 290: "*Nevertheless, it is worth noting that the extent to which the RMSE increases after shuffling SST shows a tendency to decrease as the forecast lead time increases. This result suggests that the SST itself has the strongest predictive power in forecasting SST, slightly losing predictive skill increasing the lead times. The incoming solar radiation, to a lower extent, shows the opposite behaviour: after shuffling, the RMSE tends to increase more than the other drivers with the forecast lead times. The RMSE at L7 reached values of about 1°C, 1.7°C, 1°C for WM, CM and EM, respectively. Surprisingly, in contrast with the mutual information analysis, we can notice that for WM and EM the latent heat plays a role. In particular, for WM the RMSE at L7 reached values of about 0.5 for all the ML techniques, double of the RMSE of REXPs. Overall, the aforementioned analysis suggests that the incoming solar radiation, as shown also by the mutual information analysis, has some predictive power in driving SST variability. It is important to highlight that incoming solar radiation shows a tendency to gain predictive power as forecast leads increase, whereas SST, to a much lesser extent, tends to lose it. This suggests that atmospheric variables could be useful in forecasting SST at longer time scales. Nevertheless, it is worth mentioning that the ML methods look for statistical relations (e.g. linear or non-linear relations) between variables that do not necessarily have a physical meaning (e.g. a cause-effect relation).*"

[Figure]

Figure 2. SST Root Mean Square Error (RMSE) of the sensitivity experiments SEXPs for each forecast lead time for: (left column) Western Mediterranean, (middle column) Central Mediterranean, (right column) Eastern Mediterranean. The labels indicate, for each experiment, the driver that has been shuffled. Right y axis (orange ticks) refers to SST driver experiment (orange solid line with square marker), left y axis indicates the other drivers error.

**9) Discussion and conclusions:**

**Can you introduce possible future developments?**

● **L278-282: All the abbreviations are already introduced**

● **L299-301: These features are typical of all the NNs methods. I would prefer to have this general info in the Introduction section.**

● **L330-336: Too much information on the impact, I suggest reducing this paragraph.**

Thank you for your comments, but we opt to reintroduce the abbreviation in the Discussion section for readability purposes, particularly in this crucial part of the manuscript. Additionally, we choose to retain the impacts paragraph as we believe this information is pivotal for comprehending the objectives of our work, following reviewer1's observation. However, we plan to relocate the general information about NN methods to the introduction at line 35: "*As a "learning from data" approach, machine learning has the advantages of computational efficiency, accuracy, transferability, flexibility and ease-of-use in ocean forecasting studies (Boukabara et al., 2019; Li et al., 2020; Wei and Guan, 2022; Taylor and Feng, 2022).*"

**10) Supplementary:**
**Correct the description of Figure S4**
**Figure S4: As Figure S4 but for forecast lead time 3.**

Thank you, we corrected it.

**REFERENCES:**

Boukabara, S.-A., Krasnopolsky, V., Stewart, J. Q., Maddy, E. S., Shahroudi, N., and Hoffman, R. N.: Leveraging modern arti cial intel- ligence for remote sensing and NWP: Bene ts and challenges, Bulletin of the American Meteorological Society, 100, ES473–ES491, 2019.

[revised manuscript text omitted]

---

## Author Comment (AC2)

Dear Referee,

we would like to thank you for the careful reading of the manuscript and the constructive comments that substantially helped to improve and clarify the paper. Answers to all your comments are detailed hereafter. Corrections to the English grammar were adopted in the revised version of the manuscript according to the reviewer's recommendations, but are not reported or discussed here. All authors agree with the modifications made to the manuscript. The comments by the referee are reported in bold followed by our response (in blue). The text added to the revised manuscript is reported in italic font. The line numbers reported in the answers referred to the revised manuscript. The revised manuscript that includes track changes is also provided in pdf format.

In the following answers, we use 'Figure' to identify the figures in the updated manuscript and we use 'Plot' to identify the figures in this document.

**General comments**

**1) First, the construction of the ML classification for extreme events. By definition, an extreme event is a phenomenon that is rare in a dataset, making it challenging to identify patterns and leading to imbalanced classes. In my opinion, this situation may explains the high number of FPR you found in the MHW prediction. This question about extreme events is crucial in the context of climate variations driven by climate change. The data used in this study seem not to be detrended, allowing extremes to originate from different sources, even when only meteorological predictors are considered. It would be beneficial to have a smaller sample in the training dataset with an acceptable bias to provide the network with more information about extremes (it a common way of doing, for example you can refer to the work by Mounier et al. 2022).**

**Mounier, A., Raynaud, L., Rottner, L., Plu, M., Arbogast, P., Kreitz, M., ... & Touzé, B. (2022). Detection of bow echoes in kilometer-scale forecasts using a convolutional neural network. Artificial Intelligence for the Earth Systems, 1(2), e210010**

Thank you for your insights regarding the construction of our ML classification model for extreme events, especially the challenge of identifying rare phenomena within imbalanced classes.

We acknowledge the significance of extreme events, particularly in the context of climate change-induced variations. To address this concern, we attempted to integrate Marine Heatwaves (MHWs) into the training dataset by substituting periods without MHWs with periods of MHWs. In other words, in the new training dataset we replicated randomly days of MHWs, substituting them to no-MHWs days. We did the same for the predictors, which means that we substituted the values of the predictor during the MHWs days replicated in the new dataset. Despite this effort, as depicted in Plot 1 and 2, this substitution did not notably enhance the forecasting performance for SST or MHWs. LSTM identifies the "standard" experiment, with non-inclusion of additional MHWs in the training dataset and evaluated in the 2017-2021 period of testing (i.e. REXP, see manuscript). LSTM_S is the same experiment, but trained by the new

training dataset which includes more MHWs. In addition, also the F1 score worsens substantially (Plot 2), likely linked to the fact that the model tends to overestimate maxima leading to a large number of False Positive in the LSTM_S experiment (Plot 3).

It is important to underline that in both experiments the RMSE error is calculated against the observed ESA CCI SST from 2017-2021 without synthetic (i.e. additional MWs) events.

The suggestion of the reviewer to mirror the methodology utilized by Mounier et al. 2022, by creating smaller samples with acceptable bias to provide the network more information about extremes, is a challenge for our type of framework. Temporal constraints and seasonality within the dataset made it tricky to purely eliminate days without MHWs without disrupting the dataset's temporal continuity.

Our attempt to enrich the training dataset with extreme events did not yield substantial improvements in forecast accuracy for SST or MHWs. Therefore, after extensive evaluation and considering the constraints regarding the elimination of periods without MHWs, we made the decision to continue using the original dataset without synthetic additions.

[Figure]

Plot 1. LSTM and LSTM_S performance for the SST daily predictions in terms of Root Mean Square Error (RMSE) for Western Mediterranean (Region 4).

[Figure]

Plot 2. LSTM and LSTM_S F1 score for the SST daily predictions in terms of Root Mean Square Error (RMSE) for Western Mediterranean (Region 4).

[Figure]

Plot 3. LSTM and LSTM_S False Positive Rate for the SST daily predictions in terms of Root Mean Square Error (RMSE) for Western Mediterranean (Region 4).

**2) The second concern relates to the study's objectives. If I read correctly, the justification of this study is the need for a prediction system for SST and MHW due to their significant social, ecological, and economic impacts: I couldn't agree more. However, I did not find any justification, and I'm not convinced by, the impact of having a short-term forecast of SST/MHW. Given the profound impact of MHWs on marine ecosystems and, consequently fisheries, and the timescales of MHWs (lasting for days to years) many researchers are focusing more on multi-weeks or seasonal forecasts to anticipate these impacts. I can speculate on why you developed this short-term solution, but it is not clearly justified in the article and seems to be useful for very specific industry. For example, does an extreme 1week MHW has more impact than a moderate MHWs that lasts 1 year.**

Thank you for your thorough review and insightful comments regarding our study's objectives and the justification for short-term forecasts of SST and MHWs.

The profound impact of MHWs on marine ecosystems, fisheries, and associated socio-economic factors is indeed a critical concern (Oliver et al., 2021). We recognize the merit of longer-term forecasting for understanding and mitigating lasting impacts, but short-term forecasts supply useful information in various sectors, specifically in industries reliant on immediate responses to temperature fluctuations (DeMott et al., 2021).

Short-term SST and MHW forecasts are particularly relevant in sectors such as aquaculture, where the acute impacts of short-duration extreme events can significantly impact operations (DeMott et al., 2021, Froehlich et al., 2018). For instance, an extreme MHW lasting a week might impose sudden stresses on temperature-sensitive aquatic species (e.g. the early life stages of fish, Buttay et al 2023), potentially causing immediate adverse effects compared to a more prolonged, moderate MHW (e.g mass mortality, Lee et al 2018, Buttay et al 2023). Timely predictions of these short-term fluctuations enable proactive management strategies (Guinaldo et al. 2023), aiding in disease prevention, optimizing feeding schedules,

potential coral bleaching events prevention, and ensuring the well-being of farmed species, hence safeguarding economic interests and sustaining food security (Oidtmann et al. 2011, DeMott et al., 2021).

We recognize the necessity of clearly justifying the choice of short-term forecasting objectives in our manuscript. In the revised version, we will explicitly discuss the relevance and specific impacts of short-term MHWs, particularly in sectors such as aquaculture, elucidating their importance in mitigating acute risks and ensuring the resilience of these industries.

We appreciate your valuable insights, and we elaborate on these points comprehensively in the revised manuscript. In particular in the introduction at lines 59: "*In short, given the impacts of MHWs on ocean ecosystems and the resulting economic losses of marine industries, there is an increasing need for MHW forecast to help ocean users to be prepared for these events. Short-duration extreme marine heatwaves (MHWs), lasting only a week, can exert sudden stresses on temperature-sensitive aquatic life stages, potentially leading to immediate adverse effects such as mass mortality, disease, large-scale coral bleaching and reduced seagrass meadows (Holbrook et al., 2020). For proactive marine management, operators in the coastal and marine sectors (e.g. fisheries and aquaculture, and coastal water management) can use MHW event predictions for better planning of their activities.*", and in the Discussion section at lines 373: "*We recognize the merit of longer-term forecasting for understanding and mitigating lasting impacts, but short-term forecasts supply useful information in various sectors, specifically in industries reliant on immediate responses to temperature fluctuations (DeMott et al., 2021). Short-term SST and MHW forecasts are particularly relevant in sectors such as aquaculture, a very important activity in the Mediterranean sea, where the acute impacts of short-duration extreme events can significantly impact operations (DeMott et al., 2021, Frölicher and Laufkötter 2018). For instance, an extreme MHW lasting a week might impose sudden stresses on temperature-sensitive aquatic species (e.g. the early life stages of fish, Buttay et al. 2023), potentially causing immediate adverse effects, such as mass mortality (Lee et al., 2018; Buttay et al., 2023) compared to a more prolonged, moderate MHW. Timely predictions of these short-term fluctuations enable proactive management strategies, aiding in disease prevention, optimizing feeding schedules, potential coral bleaching events prevention, and ensuring the well-being of farmed species, hence safeguarding economic interests and sustaining food security (Oidtmann et al. 2011, DeMott et al. 2021)*".

**3) In connection with this, I would suggest exploring a comparison between these ML techniques and multiple linear regression using the same predictors. This would help justify choosing ML over a statistical model with lower computational costs, as you mentioned in Page 2, Line 42.**

We appreciate the opportunity to further clarify our methodology and choices regarding the comparison between machine learning (ML) techniques and multiple linear regression (LR).

In response to your request about comparing ML techniques with LR using the same predictors, our comprehensive evaluations revealed insights, particularly regarding the impact of spatial resolution on performance. When examining the averaged timeseries of SST and predictors within a single area, LR's performance appeared comparable to that of LSTM (Plot 4). Indeed the LR error falls within the margin of error, as you noticed for MEdFS.

Nevertheless, investigating the LSTM performance in forecasting 7 days of SST for individual grid points within the same area (e.g. Area 4) revealed distinct performance differences. In this analysis we analyzed 10% of the grid points in the area to be statistically relevant. The two methods are trained using 1981-2016 as the training period and 2017-2021 as the testing period (i.e. REXPs). Solid lines, which identifies the mean of the RMSE of all the randomly selected grid points, demonstrated that LSTM consistently outperformed LR, particularly at lead times greater than 2 days (Plot 5). This highlights the sensitivity of forecasting models to the spatial resolution of the data used. As noted in the review comment number 9 (below), the averaged data may potentially smooth out localized variations, thereby potentially hiding the non-linearities between variables. ML techniques, unlike LR, are designed to identify such nonlinearities. Therefore, it aligns to find the LSTM to outperform LR when operating at a higher spatial resolution.

It's crucial to note that our aim is to compare how well ML techniques perform against a short term dynamical forecast model (MedFS) in terms of average forecast skill in selected 16 areas ad particularly in the Algerian basin, North Adriatic Sea and Southern Levantine Sea, which are identified to be amongst the most impacted areas in terms of occurrence, duration or intensity of marine heat waves (Dayan et al., 2023; Pastor and Khodayar, 2023). These areas, categorized based on their dynamics, in the context of Copernicus Marine Service are operationally provided with a forecast skill. Our manuscript primarily focuses on demonstrating a proof-of-concept for developing short-term forecasting systems. As these systems will likely be used in smaller areas or with higher-resolution data in the future (e.g. aquaculture sites), we decided to leave out LR from the paper. This decision reflects the fact that our work serves as an initial step towards developing more sophisticated systems capable of capturing complex, non-linear relationships.

We recognize the necessity of addressing limitations which are derived from the spatial resolution of our data. Therefore, we plan to incorporate it in the Discussion section in the revised manuscript, discussing in depth the influences of spatial resolution on model performance and broader implications for future research.

[Figure]

Plot 4. LSTM and LR performance for the SST daily predictions in terms of Root Mean Square Error (RMSE) for Western Mediterranean (Region 4) using the averaged time-series.

[Figure]

Plot 5. LSTM and LR performance for the SST daily predictions in terms of Root Mean Square Error (RMSE) for Western Mediterranean (Region 4) using the averaged time-series. Shading areas are the respective standard deviations.

**Abstract**
**4) Regarding your results, you could not conclusively state that ML techniques 'outperform' MedFS, even in the context of 3-day predictions. In my opinion, the results fall within the margin of error (please refer to my comments in the Results section). Furthermore, you should replace 'most regions' with a quantitative assessment of this outcome. I also suggest adding a closing sentence to your abstract to provide some perspective. However, the decision is up to you.**

Thank you for your comments, we rephrased the abstract:
"*Marine heatwaves (MHWs) have significant social and ecological impacts, necessitating the prediction of these extreme events to prevent and mitigate their*

*negative consequences and provide valuable information to decision-makers about MHW-related risks. In this study, machine learning (ML) techniques are applied to predict Sea Surface Temperature (SST) time series and Marine Heatwaves in 16 regions of the Mediterranean Sea. ML algorithms, including Random Forest (RForest), Long short-term memory (LSTM), and Convolutional Neural Network (CNN), are used to create competitive predictive tools for SST. The ML models are designed to forecast SST and MHWs up to 7 days ahead. For each area, we performed 15 different experiments for ML techniques, modifying the training and testing periods. Alongside SST, other relevant atmospheric variables are utilized as potential predictors of MHWs. Datasets from the European Space Agency Climate Change Initiative (ESA CCI SST) v2.1 and the European Centre for Medium-Range Weather Forecasts (ECMWF) ERA5 reanalysis from 1981 to 2021 are used to train and test the ML techniques. For each area the results show that all the ML methods performed with minimum Root Mean Square Errors (RMSE) of about 0.1°C at a 1-day lead time and maximum values of about 0.8°C at a 7-day lead time. In all regions, both the RForest and LSTM models consistently outperformed the CNN model across all lead times. LSTM has the highest predictive skill in 11 regions at all lead times. Importantly, the ML techniques show results similar to the dynamical Copernicus Mediterranean Forecasting System (MedFS) for both SST and MHW forecasts, especially in the early forecast days. For MHW forecasting, ML methods compare favorably with MedFS up to 3-day lead time in 14 regions, while MedFS shows superior skill at 5-day lead time in 9 out of 16 regions. All methods predict the occurrence of MHWs with a confidence level greater than 50% in each region. Additionally, the study highlights the importance of incoming solar radiation as a significant predictor of SST variability along with SST itself.*"

**Introduction**

**5) As mentioned in the general comments, the introduction lacks justification for considering only a 7- day forecast period. If we compare this to other scientific fields, such as numerical weather prediction, the justification for using ML techniques often leans in the opposite direction: attempting to outperform models beyond the 7-day threshold, where predictability reaches its limits.**
**In my opinion, this approach would be even more compelling for SSTs, given the relatively small range of variation within a week, especially in your case when utilizing interpolated L4 data for training.**

Thank you for your comment, we addressed this comment in the general comments section.

P1.L18-21
**6) I would recommend rewriting the introduction to separate the sections addressing SST, the impact on ecosystems, and the definition of MHWs as extreme events. In my opinion, there is a bit of confusion (at least as a reader) between these concepts in the initial part of the introduction, and you are defining them a second times in the Results section (Page 8, Line 229).**
**7) I think you could shorten this section which is currently a very long review. As you mentioned, statistical techniques have a well established history in SST including bias estimation and satellite data reconstruction.**

Thanks for your input. We've revised the introduction based on your comments [6] and [7], as well as those from reviewer 2.

P2.L39
**7) I'm not familiar with the reference you mentioned but I have in mind one counter example. In numerical weather prediction AI models like FourCastNest, the model provides forecasts globally for up to 15 days using ML techniques exclusively. This raises an important questions about the applicability of your study, especially given the wide area you are examining. Following this sentence it would be preferable to use numerical approach in the case of Med Sea and I could not find the justification in your introduction.**

Thank you for highlighting this point. There was a misinterpretation in our previous statement. We initially perceived "location-specific" studies as pertaining to time-series analyses. Consequently, we've removed this sentence from the introduction to avoid confusion.

**Methodological Framework**
**9) One of my main concerns in this section deals with the choice of the data and its inherent justification. If I understand correctly, the aim of the study is to develop a SST forecast up to 7 days. However, L4 ESA CCI SST product is primarily designed for climate studies and is a gap-free product, meaning that some of the data are computed using interpolation. By using such data, you may inadvertently introduce biases into your model. For example, data are smoothed and you may hide upwelling in coastal areas. An alternative approach could have been to use dedicated L3 operational data (near real time product), such as those developed by the same ESA CCI project or the OSI SAF project.**
**In connection with this, how do you account for the differences in spatial resolution between the ESA CCI and ERA5 datasets? You are attempting to link SST variations to atmospheric predictors but ERA5 has a spatial resolution of 0.25° whereas ESA CCI provides SST products at a resolution of 0.05°. I might have missed something in the study but it appears to me that you are calculating a mean SST for each area. Do you have a single SST representative for each zone? I have the same question with the atmospheric variables. However these zones represent large areas and by aggregating data in this way you may smooth out the data. Thereby by neglecting the non-linearity of wind speed, which is a key driver of SST variations especially in the Mediterranean Sea where you have regional winds that interact strongly with SSTs.**

Thank you for your insightful observations regarding our choice of data and its potential impact on our study's outcomes. Your points about the L4 ESA CCI SST product being primarily designed for climate studies and its gap-free nature are well-taken. Indeed, the gap-free nature of this dataset, relying on interpolation methods, might introduce certain biases and inadvertently smooth out potentially localized phenomena such as upwelling in coastal areas.

In our pursuit to develop a proof of concept for SST forecasting up to 7 days, ensuring gap-free data was essential to guarantee continuity and reliability in our analysis. We

agree that while the ESA CCI SST product serves this purpose, it might introduce certain limitations, as you rightfully pointed out.

Regarding the spatial resolution discrepancies between the ESA CCI and ERA5 datasets, our methodology involves averaging SST data to obtain a single representative SST for each zone and the same approach is used for atmospheric predictors. We acknowledge that aggregating data in this manner might potentially smooth out localized variations, thereby potentially overlooking the non-linearities of variables such as wind speed, particularly in regions like the Mediterranean Sea with strong interactions between regional winds and SSTs.

We aim to include these crucial considerations in our Discussion section, acknowledging the potential biases introduced by the gap-free nature of the ESA CCI SST product and the limitations arising from spatial resolution discrepancies between datasets. While our current approach facilitates gap-free data, we recognize the need for further exploration and considerations for potential refinements in future studies, particularly those focusing on more localized phenomena.

We added at lines 316: " *We acknowledge that averaging data in this manner might smooth out localized variations, thereby potentially overlooking the non-linearities of variables such as wind speed. When data are averaged within an area, the issue tends to become more linear, allowing simpler methods like linear regression to yield good results (not shown). However, when working with higher resolution data, ML methods outperform classical methods demonstrating their potential and unearthing intricate nonlinear relationships.* "

**11) I'm also surprised that you did not account for 2m air surface temperature, specific humidity and mixed layer depth variables as they are known drivers of MHWs and even more in the Mediterranean Sea. You took T2m and q2m into account in the sensible/latent heat fluxes but it would be interesting to know about their direct impact.**

Thank you for highlighting the potential impact of variables like 2m air surface temperature, specific humidity, and mixed layer depth as drivers of Marine Heatwaves (MHWs), particularly in the Mediterranean Sea.

In response to your query, we indeed incorporated 2m air temperature (T2m) and specific humidity (q2m) from the ERA5 dataset by substituting them into the computation of sensible and latent heat fluxes for region 11, for example (Plot 6). As indicated in the attached figure, our analysis revealed that incorporating these variables directly (T2Q2) did not yield a notable advantage over using the sensible and latent heat fluxes (HEATs) in predicting SST.

Despite the recognized importance of variables like T2m and q2m, in influencing MHWs, our findings did not demonstrate a significant improvement in predictive capability when utilizing these variables directly compared to their indirect incorporation through heat fluxes. This suggests that the direct inclusion of T2m and q2m did not offer a substantial enhancement over the information already encapsulated within the sensible and latent heat fluxes for the selected regions.

We greatly appreciate your insightful suggestion and the opportunity to explore the direct impact of these variables. The absence of a discernible advantage in using T2m and q2m directly, as demonstrated in our analysis, will be appropriately reflected in the revised manuscript. It underscores the importance of comprehensive assessments to discern the most influential predictors for MHW forecasting, especially in regions like the Mediterranean Sea.

We address the consideration about the MLD in the following comment [12], where the referee also pointed out the fact that we are considering just atmospheric forcing.

[Figure]

Plot 6: Performance of REXP for the SST daily predictions in terms of Root Mean Square Error (RMSE) for Central Mediterranean (Region 11). T2Q2 is the experiment with T2m and q2m drivers, HEATs used instead the sensible and latent heats as drivers.

**12) I'm also referring to the MLD because MHWs often have a vertical extension and can persist in intermediate layers without necessarily having any signature at the surface. One of the advantages of the numerical approach is its ability to account for a wide variety of processes that are parameterized within the model. However, here you are only exclusively studying SST predictions within the scope of the atmospheric forcings. How do you account for oceanic processes? Given their potential significance? For example, smoothing wind speed variations might reduce the contribution of the vertical mixing among other crucial oceanic processes. You mention about this in your discussion but this could be extended.**

Thank you for highlighting the importance of considering oceanic processes, specifically the Mixed Layer Depth (MLD), in the context of forecasting Marine Heatwaves (MHWs). We acknowledge the significance of MLD, especially regarding the vertical extension and persistence of MHWs in intermediate layers without distinct surface signatures.

In our study, we deliberately focused on surface variables, particularly Sea Surface Temperature (SST), driven by atmospheric forcings, due to their primary role in the onset of MHWs within short-term forecasting horizons. This emphasis allowed us to

prioritize the immediate drivers responsible for MHW initiation and development (Oliver et al., 2018).

Regarding the exclusion of MLD and other oceanic processes, which indeed play crucial roles in MHW evolution, we plan to thoroughly address these criticalities in the "Discussion" section of the manuscript. We aim to expand upon the discussion, acknowledging the importance of these oceanic factors and their potential influence on MHW forecasts.

We added at lines 379: "*Our study predominantly focuses on the influence of atmospheric forcings on SST predictions, motivated by their significant role in the onset of MHWs (Schlegel et al. 2021, Darmaraki et al. 2023). While this approach captures pivotal aspects, it fails to account for a variety of oceanic processes that could contribute significantly to SST variations and MHWs events. The drivers of MHWs are currently not fully understood (Hoolbrook et al. 2019), and the relevant physical drivers and processes involved in MHW emergence span various timescales, ranging from days (e.g., anomalous heat fluxes) to weeks (e.g., blocking systems and atmospheric teleconnections), months (e.g., re-emergence of warm anomalies from the subsurface), and years (e.g., climate modes and oceanic teleconnections). By omitting these factors, our study might provide an incomplete understanding of the intricate interplay between oceanic and atmospheric dynamics, limiting the comprehensiveness of our SST forecasting model. Introducing additional ocean variables alongside atmospheric data could enrich the predictive capacity of ML techniques, improving predictions by integrating a more comprehensive set of influencing factors.*"

**13) Regarding the detection of MHWs detection, the methodology is not entirely clear for me. Firstly, I could not find any explanation of how you calculate the daily climatology and the seasonal threshold. Are these values unique for each regions or are they calculated for every pixels at the ESA CCI resolution? Do you consider an MHW to occur when at least 1 pixels within a region exceeds the threshold or do you consider the mean SST over the region? Furthermore, calculating a climatology by only taking the mean of daily SSTs over 30 years can introduce some biases because it may not be statistically robust and sensitive to extremes. Typically the climatology is computed for a specific day using 30 years of data either with an 11-day sliding window centered on the day in question or by employing the first harmonic of a Fourier series.**

**Do you account for situations when a MHWs occurs less than 2 days before the end of another one it is considered part of the MHW? P6.L175: you mention computing the climatology over the period 1981-2016 which is not the international standard of using 30 years. It is worthy noting that the first complete year of ESA CCI data is 1982. More generally, you should clarify how you calculate the climatology and anomalies.**

Thanks for pointing that out. We've incorporated more details in the Methodology section. At lines 98 we stated "*SST and AtmV are averaged over the Mediterranean regions (Figure 1) to obtain time-series of SST and AtmV from 1981 to 2021.*" Our decision is to calculate climatology using all available years, and we've provided further clarity on this choice in lines 173 "*Notably, the test time-series represents the*

*regional mean SST, resulting in a single time-series per region. MHWs are defined in each regional time-series as in Hobday et al. 2016: SST higher, for 5-days or longer, than the 90th-percentile threshold of seasonally varying climatology calculated over more than 30-year period without removing long-term trend. If an event occurs less than 2 days before the end of another, it is considered as part of the ongoing MHW. An 11-day sliding window centered on each day is used for the climatology calculation. The climatology and the threshold are calculated over the reference testing period (i.e. 1982-2016) and applied to the test period (i.e. 2017-2021).... "* .

**14) In studying MHWs, it is important not only to detect them but also to estimate key metrics such as the mean intensity, max intensity and the severity. Do you have insights into the possibility of predicting these metrics using your model?**

Thanks for your observation. We opted to evaluate the mean forecasted intensity (I_mean_ML) within the REXPs for each ML technique against the observed intensity mean from the ESA CCI dataset (Imeanesa). This comparison allowed us to compute the Intensity Error (IE), calculated as IE = I_mean_ESA - I_mean_ML. Figure 1 has been incorporated into the supplementary material, and the outcomes for the region 4-11-15 are additionally presented in Figure 4 of the manuscript. To elaborate on this figure, we've included corresponding commentary at lines 277. "*We also evaluate the differences in MHWs intensity mean predicted by the ML models and MedFS against observation (i.e. intensity error, IE, see "Evaluation metrics" section) during the studied period (Figure 4c). For WM, CM, and EM it is interesting to note that MedFS shows positive IEs, meaning that the MedFS predicted intensity is overestimated, while the ML models show opposite behavior, IEs are negatives. This characteristic is evident also for the other regions (Figure S6), except regions 1, 12, 13 where MedFS IEs are negative. It is also worth highlighting that, in all the regions, the RForest IEs, even if they are not the smallest ones, degrade slower than the LSTM IEs.* "

[Figure]

Figure 1. Forecasted MHWs mean intensity error (IE) as a function of the forecast lead time for each region.

**P4.L104**

**15) This part should be moved to the results and discussions sections. Moreover, GEO and INC are not independent variables, which may explains why they are correlated with SST. Even though you are looking at daily mean, ESA CCI SST are representative of the foundation temperature which is not influenced by the diurnal cycle. Therefore, you should rephrase the sentence 'INC influenced directly SST during daily time' (P4.L7). I understand what you meant and suggest this modification to avoid the reader to misunderstood.**

Thanks for your feedback. We relocated this segment to the Results section and revised the sentence as lines 204.

**16) I'm not entirely convinced that MI calculations are necessary to identify relations between variables. Examining the upper-ocean layer equation could give the same result (although it is not quantitative). Additionally, it is excepted to find a weak correlation between wind speed and SST due to the non-linearity I mentioned earlier which you did not take into account.**

Thank you for your thoughtful considerations regarding the necessity of mutual information (MI) calculations in identifying relationships between variables and the potential impact of non-linearities, particularly in the context of wind speed and Sea Surface Temperature (SST).

While examining equations, such as those governing the upper-ocean layer, can indeed provide insights into variable relationships, measures like MI calculations offer a valuable understanding of associations and dependencies within the data (Kraskov et al., 2004). MI provides a rigorous and data-driven approach to quantifying dependencies that might not be readily apparent through qualitative examinations alone. In addition, differently from the correlation coefficient, MI is a general measure of association able to account also for non-linear dependencies. In our proof-of-concept study, utilizing MI calculations enabled us to identify significant associations between variables, offering insights into potential drivers that might influence SST predictions based on the available data.

This proof-of-concept study aimed to demonstrate an approach to develop forecasting systems. By leveraging MI calculations, we aimed to showcase a methodology for extracting insights from available data, recognizing that this is a preliminary step towards developing more sophisticated systems capable of capturing complex, non-linear relationships.

P5.L124-141
**17) I have several questions regarding the description of the networks you used. Firstly, you have not detailed the inputs/ouputs and how the variables are incorporated in the networks (possibly as concatenated large vectors?). Additionally, I found no indications in this work regarding how the data were normalized to make different atmospheric and oceanic variables comparable. Additionally, I find it somewhat surprising that you used 200 epochs for both LSTM and CNN without discussing the complexity of the networks. For instance, in atmospheric models, LSTM converge quickly compared to CNN and it is common practice to add an early stopping to prevent overfitting.**

Regarding the network descriptions, we apologize for the lack of detailed information in our initial presentation. Thanks to your feedback, we enhanced our networks and conducted a complete revision of all our experiments. We indeed used concatenated large vectors as inputs to our networks, structured as [time-steps, lags, variables]. The data normalization was carried out using a min-max scaler, ensuring comparability across various atmospheric and oceanic variables. We added the following text at lines 115: "*To ensure comparability across diverse atmospheric and oceanic variables, we normalize this data using a min-max scaler. These ML frameworks receive concatenated extensive vectors structured in the format [time-steps, lags, variables]. Here, 'lags' denotes the previous 7 days, while 'variables' encompass SST and the predictors*".

We appreciate your concern about potential overfitting and the complexity of the networks. We conducted a thorough experimentation, including employing dropout and regularization techniques for the LSTM model. Surprisingly, these methods did not yield improved results over the direct use of 200 epochs. In fact, we plotted the training loss versus the validation loss, and both curves exhibited similar behavior (Plot 7). They showed a rapid decrease initially, followed by a stabilization, indicating that additional epochs or complex regularization techniques did not significantly enhance the model's performance.

Our findings suggested that the model's capacity might have been well-suited to the dataset, thus avoiding overfitting despite the higher number of epochs. However, we acknowledge the importance of discussing such aspects in the methodology section to provide a clearer understanding of the network's behavior and our decision-making process. We added this sentence at lines 138: "*Notably, in our methodology, we opted not to incorporate dropout layers within the network architecture. Despite the absence of dropout regularization, our training results did not exhibit overfitting tendencies, suggesting a congruence between the model's capacity and the dataset characteristics.*"

[Figure]

Plot 7. Training (blue line) and validation (red line) loss progression over epochs during model training.

P6.L158
**17) This is a minor comment but you mentioned computational time without providing hardware specifications. I may find this information useful.**

Thank you for your comment. We corrected the computational time in "*The computational time required for the ML experiments is about 2 minutes to train the method and 30 seconds to test it*". The previously reported computational time in our manuscript encompassed data preprocessing, which included preparing the time-series. To provide clarity, we've now isolated and disclosed the actual time taken by the models.
Regarding the hardware specifications, our experiments were conducted on the CMCC supercomputer ZEUS. We've included the detailed specifications at lines 163: "*We run all the experiments on the CMCC supercomputer ZEUS, which comprises 348 Lenovo SD530 biprocessor nodes (totaling 12,528 cores) interconnected via an Infiniband EDR network and boasts a total theoretical computing power of 1,202 TFlops*"

**18) Equation 4: There is a typo in the F1 score, it is 2*P*R/(P+R)**
Thank you, we corrected it.

**Results**

**19) I'm a bit surprised by the lack of stability in the ML techniques (this might be related to a lack of experience with these techniques from my side). On the contrary, as you mentioned in P7.L210, it is normal for the dynamical model to be rather stable for up to 7 days (in my opinion it may depends on the predictability limits). Are you sure that the behavior of your ML models is not influenced by some sort of persistence or memory effect that could explain the increase in RMSE? I thought about this because you may have introduced some biases during the training stage, as I mentioned earlier. Additionally, you are also studying SST through a spatial average over large areas. Furthermore, at the first order, a 0.1°C difference is within the common measurement errors in satellite SST products. Regarding all this, I would not claim that ML techniques 'outperformed' the numerical model, but as you mentioned in P8.L213, they 'compare favorably'.**

In order to investigate the causes of the deterioration of RMSE at larger forecast leads we considered two alternative specification for the ML models, differing from the one described in the paper in the time lags for the inputs: the first alternative model includes only inputs at time lag -1 and -2, the other one includes only inputs at time lags -6 and -7.

The results obtained using LSTM in region 4 are displayed in Plot 8. As expected, excluding the less recent time lags (LSTMINI) has a negligible effect on RMSE, while discarding the more recent time lags (LSTMEND) has a dramatic impact on RMSE.

Furthermore, the deterioration in RMSE is less evident with LSTMEND. In other words, the RMSE shows a flatter pattern for LSTMEND than for LSTM and LSTMINI. More specifically, there is an approximate 30% increase in the RMSE of LSTMEND from forecast lead time 1 to forecast lead time 7, compared with an approximate 200% increase in the original RMSE (and in the RMSE of LSTMINI).

It is also worth noting that the RMSE of LSTMEND at leading time 1 is comparable with the RMSE of LSTM at leading time 6, which is coherent with the implicit time distance between forecast lead time and time lags in the inputs (7 days in both cases). All these findings suggest that the short-term dependence structure (conveyed by the most recent time lags) plays a prominent role in determining the predictive ability of the ML methods considered in this paper. Clearly, this short-term dependence structure can be effective in prediction at short forecast leads, but it becomes less effective at large forecast leads.

This seems coherent with the fact that, according to the sensitivity experiments, the input with the strongest predictive power is SST (at lagged times) and, as displayed in Figure S1, the mutual information between SST and itself at different time lags tends to decrease as the time lag increases.

[Figure]

Plot 8. LSTM, LSTMINI (taking only 1st and 2nd days of lag) and LSTMEND (taking only 6th and 7th days of lag) performance for the SST daily predictions in terms of Root Mean Square Error (RMSE) for Western Mediterranean (Region 4).

**20) In a second step, your study and the results are quite interesting. In my opinion, it could be improved by an in-depth analysis of the results by regions, attempting to understand the diversity of responses from different basins (explaining it through dynamic conditions or other factors). For example, the dynamic in regions like the Alboran Sea may explain most of the variability. I understand that it may not be feasible to include everything in a single study, but this could be explored in the context of an additional study.**

Following your suggestion, we decided to plot the mean RMSE error for the LSTM REXPs experiment for each region (Figure 2). We can notice that the areas with larger errors are: (i) the Alboran Sea strongly, characterized by a complex dynamics influenced by the incoming cold Atlantic water through the Gibraltar Strait modulating the water transport; (ii) the North-West part of the basin, which is an area of dense water formation and intense dynamics due to the Gulf of Lion gyre (Madec et al., 1991; Pinardi et al., 2006) and by a boundary intense current called the Liguro-Provenal-Catalan Current (Pinardi et al., 2006); (iii) the Adriatic Sea, especially in its northern shelf, characterized by a complex topography, intense air-sea exchanges, large riverine inputs that contribute to enrich the dynamics of the area. Thus the correct SST representation and forecast in these areas represent a challenging modeling problem. We added Figure 2 as Figure 4 in the manuscript and reported this description at lines 219.

[Figure]

Figure 2. LSTM mean RMSE for REXP for each region

**21) Figure 5 is very informative, would it be possible for you to add the daily climatological mean in addition to the threshold.**

Thanks for your observation. In addition to the daily climatology we've incorporated also the annual mean and standard deviation, considering the consistently high correlation, always exceeding 0.99 (Figure 3). The Figure 3 below is now the Figure 6 in the manuscript. Additionally, Figures for lead times 1 and 3 have been included in the supplementary material. To further elaborate on these updates, we've included corresponding commentary at lines 246. "*The figure shows a very close match between the forecasts and the observations, being the SST variability clearly well represented and forecasted by all the models (i.e. ML techniques and MedFS model). This is confirmed by the fact that the difference in annual means and standard deviations are minimal, generally within decimal values.*"

[Figure]

Figure 3: Time series of observed SST (ESA CCI SST) and predicted SST by the ML techniques (LSTM, CNN, RForest) and by MedFS at 5 days forecast lead time during 2020 for: (top panel) Western Mediterranean, (middle panel) Central Mediterranean and (bottom panel) Eastern Mediterranean. The 90th percentile threshold to define MHWs is represented in gray and the daily climatology in green. Crosses correspond to missed alarms (False Negative) and points to false alarms (False Positive) in the forecasts output in predicting MHWs. Colors refer to the different ML techniques

**22)  Figure 6: I found it a bit difficult to read the figure with two distinct labels for SST and the other variables. In addition, the lines are very thin, and I can't distinguish easily between the variables.**

Thank you, we simplified  the Figure.  The Figure 4 below is now the Figure 7 in the manuscript.  It is worth noting that the values are different from the first version of the

paper. As said in comment 17, we enhanced our networks and conducted a revision of all our experiments. We explained it at lines 287:

"*The labels of Figure 7 indicate, for each experiment, the driver that has been shuffled in REXPs. For an easier interpretation of the results we are showing only the SST, the INC and the LAT drivers, as they are the most relevant.* ". We comment on this Figure at lines 290: "*Nevertheless, it is worth noting that the extent to which the RMSE increases after shuffling SST shows a tendency to decrease as the forecast lead time increases. This result suggests that the SST itself has the strongest predictive power in forecasting SST, slightly losing predictive skill increasing the lead times. The incoming solar radiation, to a lower extent, shows the opposite behaviour: after shuffling, the RMSE tends to increase more than the other drivers with the forecast lead times. The RMSE at L7 reached values of about 1°C, 1.7°C, 1°C for WM, CM and EM, respectively. Surprisingly, in contrast with the mutual information analysis, we can notice that for WM and EM the latent heat plays a role. In particular, for WM the RMSE at L7 reached values of about 0.5 for all the ML techniques, double of the RMSE of REXPs. Overall, the aforementioned analysis suggests that the incoming solar radiation, as shown also by the mutual information analysis, has some predictive power in driving SST variability. It is important to highlight that incoming solar radiation shows a tendency to gain predictive power as forecast leads increase, whereas SST, to a much lesser extent, tends to lose it. This suggests that atmospheric variables could be useful in forecasting SST at longer time scales. Nevertheless, it is worth mentioning that the ML methods look for statistical relations (e.g. linear or non-linear relations) between variables that do not necessarily have a physical meaning (e.g. a cause-effect relation).*"

[Figure]

Figure 4. SST Root Mean Square Error (RMSE) of the sensitivity experiments SEXPs for each forecast lead time for: (left column) Western Mediterranean, (middle column) Central Mediterranean, (right column) Eastern Mediterranean. The labels indicate, for each experiment, the driver that has been shuffled. Right y axis (orange ticks) refers to SST driver experiment (orange solid line with square marker), left y axis indicates the other drivers error.

**Discussion/Conclusion**

In general, the discussion is good, with some very interesting points and references. I appreciate that you are not attempting to oversell your results. ML techniques, despite some limitations, have demonstrated predictive skill, and your discussion about the challenges in understanding MHW, along with a thoughtful exploration of time-scales, raised some intriguing questions.

**23) L293-295: In regard of your results the ML techniques seems to have better result however regarding the stability, I'm not convinced that it is not linked to some memory effect. I'm also not sure of the impact of predicting SSTs at 1 and 3 days, except for very specific industry with high tolerance to the risks.**

Please refer to comment 19 above.

**24) L310: I agree with the contribution of incoming solar radiation, which is generally linked to lower than average anomalies in cloud cover. However, this contribution is limited in summer in the Mediterranean Sea due to the usual low cloud cover over the area. Thus, MHWs are primarily driven by other variables, such as heat fluxes (namely the atmospheric variables T2M, Q2M and WS) which exhibit significant regional dependencies (Guinaldo et al., 2023, as mentioned earlier). Additionally, you could have also discussed the possibility of incorporating ocean heat content to improve the forecast (Holbrook et al., 2020).**

Thank you for your input. Figure 4 above indicates the significance of latent heat, particularly in Region4. Hence, we've included a sentence in the results section acknowledging the influence of latent heat: "*..we observe the impact of LAT on WM and EM..*". However, our perspective aligns with the idea that machine learning methods seek statistical correlations among variables, often without reflecting a direct cause-and-effect relationship. Regarding the ocean heat content, as addressed in comment 12, we deliberately emphasized atmospheric forcings as predictors due to their pivotal role in short-term forecasting of MHWs.
It is worth mentioning that Guinaldo et al. 2023 emphasized the need of short term observation and forecast for the detection of harmful algal blooms that might have endangered several coastal zone, shifts in community organisation (from temperate to tropical species) and prevention in mass mortalities of endemic species. Thus, we added Guinaldo et al 2023 as a reference at lines 379.

**25) Another important point to consider is related to the data you used. As mentioned earlier, you employed SST data dedicated to climate studies instead of near-real-time data. Your discussion would benefit from addressing the limitations associated with using such data. In the context of the study framework, it would also be worthy to discuss the comparison between this type of forecasting model and other approaches, such as multiple linear regression or model ensembles. To enhance this discussion, I recommend reading the following study:**

**Benthuysen, J. A., Smith, G. A., Spillman, C. M., & Steinberg, C. R. (2021). Subseasonal prediction of the 2020 Great Barrier Reef and Coral Sea marine heatwave. Environmental Research Letters, 16(12), 124050.**

Please refer to the answer 1 and 2 in the general comments.

**References:**

Buttay, L., Ohlberger, J., & Langangen, Ø. (2023). Management strategies can buffer the effect of mass mortality in early life stages of fish. *Journal of Applied Ecology*, *60*(10), 2056-2065.

Boukabara, V. Krasnopolsky, J.Q. Stewart, E.S. Maddy, N. Shahroudi, R. N. Hoffman, Leveraging modern artificial intelligence for remote sensing and nwp: benefits and challenges, Bull. Am. Meteorol. Soc. (2019).

DeMott, C., Muñoz, A., Roberts, C., Spillman, C., & Vitart, F. (2021). The benefits of better ocean weather forecasting. *Eos*, 102. https://doi.org/10.1029/2021EO210601

Dueben, P. Bauer, Challenges and design choices for global weather and climate models based on machine learning, Geosci. Model Dev. 11 (2018) 3999–4009.

Darmaraki, S., Waldman, R., Sevault, F., and Somot, S.: Dominant drivers of Past Mediterranean Marine Heatwaves, EGU General Assembly 2023, Vienna, Austria, 24–28 Apr 2023, EGU23-13986, https://doi.org/10.5194/egusphere-egu23-13986, 2023.

Froehlich, C. Y., Klanten, O. S., Hing, M. L., Dowton, M., & Wong, M. Y. (2021). Uneven declines between corals and cryptobenthic fish symbionts from multiple disturbances. *Scientific Reports*, *11*(1), 16420.

Guinaldo, T., Saux Picart, S., & Roquet, H. (2022). Response of the sea surface temperature to heatwaves during the France 2022 meteorological summer. *EGUsphere*, 1-18.

Kraskov, A., Stögbauer, H., & Grassberger, P. (2004). Estimating mutual information. *Physical review E*, *69*(6), 066138.

Lee, D. C., Won, K. M., Park, M. A., Choi, H. S., & Jung, S. H. (2018). An analysis of mass mortalities in aquaculture fish farms on the southern coast in Korea. *Ocean Policy Research*, *33*(1), 1-16

Madec, G., Delecluse, P., Crepon, M., Chartier, M., 1991. A three-dimensional numerical study of deep-water formation in the northwestern Mediterranean Sea. Journal of Physical Oceanography 21 (9), 1349–1371. http://dx.doi.org/10.1175/1520-0485(1991)021h1349:ATDNSOi2.0.CO;2.

Oidtmann, B., & Stentiford, G. D. (2011). Disease Prediction and Forecasting in Aquaculture. In Aquaculture Virology (pp. 325-344). Academic Press.

Oliver, E. C., Benthuysen, J. A., Darmaraki, S., Donat, M. G., Hobday, A. J., Holbrook, N. J., ... & Sen Gupta, A. (2021). Marine heatwaves. *Annual Review of Marine Science*, *13*, 313-342.

Pinardi, N., Zavatarelli, M., Arneri, E., Crise, A., Ravaioli, M., 2006. The physical, sedimentary and ecological structure and variability of shelf areas in the Mediterranean Sea. In: Robinson, A., Brink, K. (Eds.), THE GLOBAL The Global

Coastal Ocean: Interdisciplinary Regional Studies and Syntheses, vol. 14. Harvard University Press, The Sea (Chapter 32).

Rodrigues, R. R., Taschetto, A. S., Sen Gupta, A., and Foltz, G. R.: Common cause for severe droughts in South America and marine 440 heatwaves in the South Atlantic, Nature Geoscience, 12, 620–626, 2019

Schlegel, R. W., Oliver, E. C. J., and Chen, K.: Drivers of Marine Heatwaves in the Northwest Atlantic: The Role of Air–Sea 425 Interaction During Onset and Decline, Front. Mar. Sci., 8, 1–18, https://doi.org/10.3389/fmars.2021.627970, 2021.

---

## Referee Report (RR1)

Egusphere-2023-1847

Dear authors,
I appreciate the effort you put into answering all of my questions. I have a few very minor comments/corrections that might enhance the flow of the text (please feel free to disregard them). The manuscript can be considered accepted after the minor comments written below. I have aligned my comments with the numbering you used in your response available at https://doi.org/10.5194/egusphere-2023-1847-AC1.

4) Abstract:
- **L9**: replace *For each area* with For each region.
  (since in the manuscript, 'east-west-central Mediterranean regions' are also mentioned)
- **L10**: add here the information of the period "*modifying the training and testing periods (from .. to ..)*"

5) Introduction:
I appreciated the proposed changes. They allow the reader to have simple and concise information on the Med. Sea followed by information on the dynamical model MedFS (comment 7).

- fix the punctuation at: "The Mediterranean Sea serves our study area due to its relevance for marine management activities, For instance, over 95% of the global production of seabream and seabass comes from aquaculture, with Mediterranean countries contributing to 97% of this production. (Carvalho, N., & Guillen, J. 2021)."

6) Methodological Framework:
- L3: add 'degree' → 0.05 × 0.05 degree

7) Experiments/Evaluation metrics:
Thank you for considering my suggestion. Please take care with the brackets at L3

8) Results:
I recommend briefly adding the information provided here about the Alboran Sea, N.West, etc., with the proposed changes mentioned at comment 5)

---

## Author Response (AR2)

Dear referees,

Thank you very much for your comments; they significantly improved the paper.
Referee 1, I have incorporated your suggestions into the abstract in the new manuscript version.

Best regards,
Giulia Bonino